# *Pparg* promotes differentiation and regulates mitochondrial gene expression in bladder epithelial cells

Chang Liu[1,9], Tiffany Tate [1,9], Ekatherina Batourina[1], Steven T. Truschel[2], Steven Potter [3], Mike Adam[3], Tina Xiang[1], Martin Picard[4], Maia Reiley[1,6], Kerry Schneider[1,7], Manuel Tamargo[1], Chao Lu [5], Xiao Chen[5], Jing He[8], Hyunwoo Kim [1] & Cathy Lee Mendelsohn [1]

The urothelium is an epithelial barrier lining the bladder that protects against infection, fluid exchange and damage from toxins. The nuclear receptor *Pparg* promotes urothelial differentiation in vitro, and *Pparg* mutations are associated with bladder cancer. However, the function of Pparg in the healthy urothelium is unknown. Here we show that *Pparg* is critical in urothelial cells for mitochondrial biogenesis, cellular differentiation and regulation of inflammation in response to urinary tract infection (UTI). Superficial cells, which are critical for maintaining the urothelial barrier, fail to mature in *Pparg* mutants and basal cells undergo squamous-like differentiation. *Pparg* mutants display persistent inflammation after UTI, and Nf-KB, which is transiently activated in response to infection in the wild type urothelium, persists for months. Our observations suggest that in addition to its known roles in adipogegnesis and macrophage differentiation, that Pparg-dependent transcription plays a role in the urothelium controlling mitochondrial function development and regeneration.

[1] Department of Urology, Genetics, and Devlopment, Pathology and Cell Biology and CSCI, Columbia University, New York, NY 10032, USA. [2] Department of Cell Biology, University of Pittsburgh School of Medicine, Pittsburgh, PA 15261, USA. [3] Division of Developmental Biology, Cincinnati Children's Medical Center, Cincinnati, OH, USA. [4] Department of Psychiatry and Neurology, Columbia University, New York, NY 10032, USA. [5] Department of Genetics and Development, Columbia University, New York, NY 10032, USA. [6] Present address: Department of Surgery, Ascension/St. John Providence, 16001 West Nine Mile Road, Southfield, MI 48075, USA. [7] Present address: College of Veterinary Medicine, Cornell University, Ithaca, NY 14853, USA. [8] Present address: Department of Systems Biology, Columbia University, New York, NY 10032, USA. [9] These authors contributed equally: Chang Liu, Tiffany Tate. Correspondence and requests for materials should be addressed to C.L.M. (email: clm20@cumc.columbia.edu)

The urothelium is a stratified epithelium that extends from the renal pelvis to the bladder neck, serving as a barrier between urine and blood. The bladder urothelium has a number of unique features; it is a long-lived epithelium, with a half-life estimated to be 40 weeks or more[1], and functions as a waterproof barrier that prevents leakage during voiding, thereby protecting underlying tissue from damage and water loss. In a full bladder, luminal superficial cells (S cells) can expand to 250 μm in length and are interconnected by high-resistance tight junctions that prevent leakage under pressure[2]. S cells are long-lived, post-mitotic cells that are critical for maintaining the urothelial barrier. They are specialized for synthesis and transport of uroplakins (Upks), a family of membrane proteins that assemble into crystals that line the apical surface of the urothelium. When the bladder expands, the S cell surface area increases, mediated by specialized vesicles that transport newly formed uroplakin crystals from the Golgi to the apical membrane[3]. During voiding, the S cell surface area is reduced by endocytic vesicles that transport apical membrane to lysosomes for degradation[3].

In addition to S cells, the urothelium contains mononucleated and binucleated intermediate cells (I cells). The mononucleated progenitor population can self-renew, or undergo incomplete cytokinesis to produce a binucleated I cell with $2n/2n$ ploidy[4]. These binucleated I cells undergo a second round of endor-eplication, differentiating into S cells with $4n/4n$ ploidy[4]. There are two known sub-populations of basal cells in the urothelium. The majority (80%) are K5-basal cells that reside in the basal and suprabasal layers and are K5+/P63+/K14−. A second population, K14-basal cells (K14+/K5+/P63+), are found exclusively in the basal layer. The adult urothelium is largely quiescent, but undergoes a rapid sequence of exfoliation and regeneration in response to injury from toxic chemicals or urinary tract infection (UTI) with uropathogenic *Escherichia coli* (UPEC). When S cells die during homeostasis or after acute injury, they are replaced by I cells[5]; however, I cells are depleted after serial injury, after which K14-basal cells expand and function as a progenitor population[6].

Peroxisome proliferator-activated receptor-γ (*Pparg*) is a nuclear hormone receptor that regulates numerous cellular functions, including adipogenesis, lipid biosynthesis, energy expenditure and storage, inflammation, and differentiation[7]. *Pparg* acts in a number of tissues and cell types, including liver, adipose tissue, and macrophages[8]. In addition, *Pparg* agonists and antagonists have an effect on the ureteral urothelium differentiation in vitro[9] and in vivo[10]. Heterodimers composed of *Pparg* and nuclear receptor family member *Rxra* regulate transcription by binding to peroxisome proliferator response elements present in regulatory regions of target genes. *Pparg* can be activated by binding of natural ligands, including fatty acid metabolites, unsaturated fatty acids such as eicosanoids, and prostaglandins[11]. A number of metabolic functions are controlled by *Pparg* in association with the co-factor *Ppargc1a*, a master regulator of mitochondrial biogenesis[12]. *Pparg* also serves as an important regulator of anti-inflammatory activity, acting in part by antagonizing the nuclear factor-κB (NF-κB) pathway[13].

Mapping of the mutational landscape of muscle-invasive bladder cancers (MIBCs) together with unsupervised clustering analysis of the whole-genome expression data revealed that MIBC can be sub-categorized into luminal and basal subtypes. These subtypes are histologically distinct and display discrete sets of mutations and gene expression signatures[14–19]. These analyses reveal alterations in *PPARG* expression and signaling, suggesting that *PPARG*-dependent transcriptional regulation may be important in the etiology of urothelial carcinoma. *PPARG* copy number expansion and increased expression of *FABP4*, a direct *PPARG* transcriptional target, were detected in luminal tumors[20–22]. Activating mutations in *PPARG* and *RXRA*, a PPARG-binding partner, were

also observed in luminal MIBCs[23,24]. In addition, genetic pathways important for lipid metabolism and adipogenesis were up-regulated in patients that harbor *PPARG* gain-of-function mutations, suggesting that *PPARG* may be an important regulator of lipid metabolism in the luminal subtype of MIBCs.

The exact contribution of *PPARG* to the etiology of the basal subtype of urothelial carcinoma is less clear. *PPARG* expression is low in basal subtype tumors compared to healthy urothelium, and *PPARG* is down-regulated in Claudin-low tumors, which have basal-like features. Interestingly, genes encoding cytokines and chemokines are up-regulated in Claudin-low basal-like tumors, which may reflect unregulated NF-κB signaling due to low levels of *PPARG*[25]. Expression of PPARG and its binding partner RXRA are reduced in the squamous cell carcinoma-like (SCCL) subtype of MIBCs, which shares many features with the basal subtype, including gene expression signatures and common mutations. Transcriptional analysis of these tumors revealed down-regulation of a large cluster of genes important for lipid metabolism, many of which have *PPARG* binding sites in their regulatory regions based on in silico chromatin immunoprecipitation-sequencing analysis[26].

In this study, we use constitutive and inducible cell-type-specific Cre mouse models to study the role of *Pparg* in distinct urothelial sub-populations. We find that *Pparg* is critical in I cells and in S cells for mitochondrial biogenesis, controlling specification and differentiation of I cells and S cells during development and homeostasis. Pparg plays an independent role in basal cells, preventing squamous differentiation. Pparg is also critical during regeneration for resolving NF-κB signaling, which is transiently increased in the wild-type urothelium in response to UPEC infection, but persists in mutants for months after UTI. Together, these findings suggest that *Pparg* is essential for normal differentiation, maintenance, and regeneration of the urothelium. Understanding the link between *Pparg*, metabolic dysfunction, chronic inflammation, and aberrant urothelial differentiation may help define strategies for urothelial generation, and could improve our understanding of the molecular changes that occur during urothelial carcinoma.

## Results

**Pparg is required for urothelial development and homeostasis.** The urothelium contains sub-populations that can be identified based on combinatorial marker expression (Fig. 1a). In adults, *Pparg* is expressed throughout the urothelium, at highest levels in S cells (Fig. 1b, c; yellow arrows). *Fabp4*, a direct transcriptional target of *Pparg*, is enriched in S cells, suggesting that *Pparg* signaling is most active in the S cell sub-population (Fig. 1d; yellow arrow). In the embryonic urothelium, *Pparg* expression is first observed in I cells at E13 (Fig. 1e, purple arrow), and between E14 and E16, is present in I cells and in maturing S cells (Fig. 1f, g, purple and yellow arrows, respectively). We did not observe detectable levels of *Pparg* in basal cells during development (Fig. 1g, green arrow).

To identify *Pparg* functions in urothelial differentiation and homeostasis, we generated B6.Cg-Shhtm1(EGFP/cre)Cjt/J; *Pparg*^tm1.1Gonz mice[27,28]; hereafter referred to as *Shh*^Cre;*Pparg*^fl/fl mutants, using the *ShhCre* driver to delete *Pparg* in basal cells, I cells, and their daughters. Analysis of *Shh*^Cre;*Pparg*^fl/fl mutants at post-natal stages revealed abnormal urothelial differentiation. Krt20 and Upk are highly expressed in S cells lining the superficial layer in controls (Fig. 2a, b), but expression is low or undetectable in S cells of *Shh*^Cre;*Pparg*^fl/fl mutants (Fig. 2e, f). In addition, mutant S cells were about half the size of wild-type S cells (S cells are denoted by dotted white circles in Fig. 2b, c, f, g). S cells in both controls and *Shh*^Cre;*Pparg*^fl/fl mutants were positive

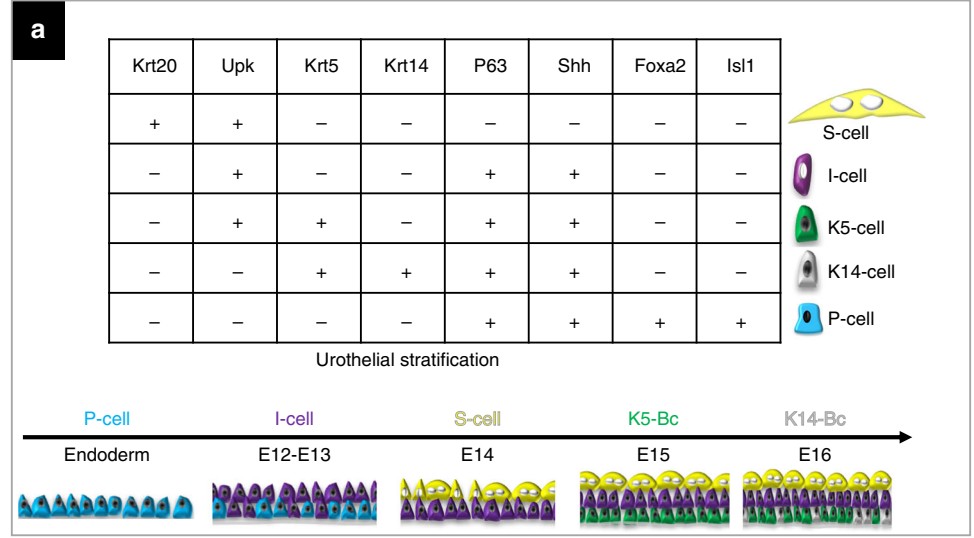

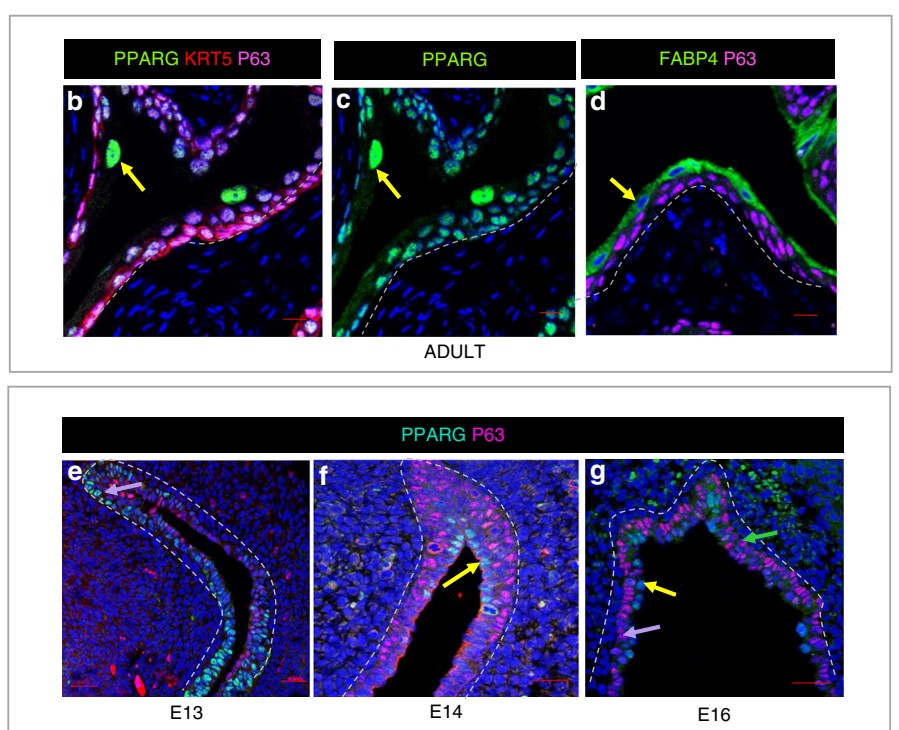

**Fig. 1** *Pparg* expression and signaling in the adult and embryonic urothelium. **a** A table showing urothelial cell types and combinatorial markers that distinguish different sub-populations. **b**, **c** Sections from a wild-type adult bladder stained for the expression of Pparg, Krt5, and P63. **d–f** Fabp4, Pparg, and p63 expression in the developing urothelium at **e** E13; **f** E14, **g** E16. Yellow arrows denote S cells, purple arrows denote I cells, and green arrows denote basal cells. Scale bar: 20 µm. Samples used in the experiments: adult wild-type urothelium, $n = 5$; E13 wild-type urothelium, $n = 4$; E14 wild-type urothelium, $n = 6$; E16 wild-type urothelium, $n = 5$

for the tight junction protein ZO1 (Fig. 2c, g); however I cells (P63+, Upk+) were not detectable in mutants at post-natal stages (Fig. 2e, f; Supplementary Fig. 1a–j), suggesting that the I cell population either failed to self-renew or regressed.

The K14-basal population, which makes up around 10% of the basal population in controls, was doubled in $Shh^{Cre};Pparg^{fl/fl}$ mutants, and K5-basal cells (P63+ Krt5+ Krt14−), which populate most of the basal and suprabasal layers in controls, were reduced in number (Fig. 2d, h–k). The expanded K14-basal cell population persisted in mutants; however, we did not observe signs of tumor formation after a year or more (Supplementary Fig. 1k–n). Taken together, these observations suggest that *Pparg*

signaling is critical for specification and differentiation of multiple urothelial sub-populations.

To learn more about the causes of urothelial abnormalities in adult $Shh^{Cre};Pparg^{fl/fl}$ mutants, we compared urothelial development in mutants and controls. Analysis at E13, when the urothelium in wild-type embryos is composed mainly of I cells, did not reveal any detectable differences in mutants compared to controls (Fig. 3a, b). At E14, the urothelium of both wild-type and mutants contains I cells (Upk+ P63+) and immature S cells (Upk+ P63-); Fig. 3b, e; I cells, purple arrows, immature S cells, yellow arrows). Analysis at E16 revealed robust Upk3a expression in S cells and I cells of controls (Fig. 3c), while in the mutants, Upk3a

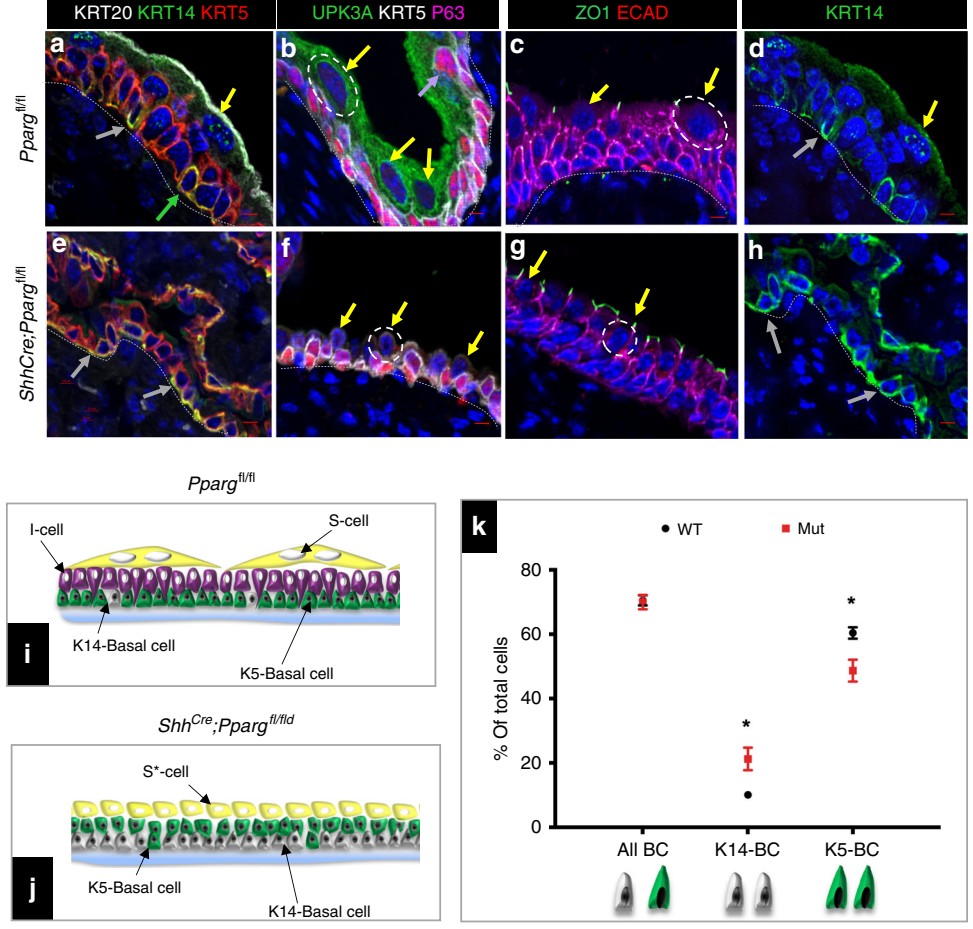

**Fig. 2** Urothelial abnormalities in adult *Shh^Cre^;Pparg^fl/fl^* mutants. **a**, **e** Expression of Krt20, Krt14, and Krt5 in an adult *Pparg^fl/fl^* control mouse (**a**) and in a *Shh^Cre^;Pparg^fl/fl^* mutant (**e**). **b**, **f** Expression of Upk3a, Krt5, and p63 in a *Pparg^fl/fl^* control mouse (**b**) and in a *Shh^Cre^;Pparg^fl/fl^* mutant (**f**). **c**, **g**)Expression of ZO1 and E-cadherin in a *Pparg^fl/fl^* control mouse (**c**) and in a *Shh^Cre^;Pparg^fl/fl^* mutant (**g**). **d**, **h** Krt14 expression in a *Pparg^fl/fl^* control (**d**) and in a *Shh^Cre^;Pparg^fl/fl^* mutant (**h**). Gray arrows denote abnormal S cells; yellow arrows denote S cells; purple arrows denote I cells; and green arrows denote basal cells. **i**, **j** A schematic representation showing cell types in the adult urothelium of a *Pparg^fl/fl^* control mouse (**i**) and in a *Shh^Cre^;Pparg^fl/fl^* mutant (**j**). **k** Quantification of numbers of Krt14 and Krt5 expressing basal cells in the urothelium of adult *Pparg^fl/fl^* control mice and *Shh^Cre^;Pparg^fl/fl^* mutants. Significance calculated by a two-tailed Student's *t* test; *$p < 0.05$. Numbers are means of percentages ± SEM. Scale bar: 50 μm. The number of animals used in this experiment was: adult mutant, $n = 9$; adult control, $n = 6$. Source data are provided as a Source Data file

was down-regulated, suggesting that *Pparg* is important for maintaining S cell specification (Fig. 3f; white arrow). We did not observe abnormalities in the basal population during development (Fig. 3b, c, e, f). Taken together, these observations suggest that *Pparg* is important for differentiation of I cells and S cells during development, and in adults, it regulates differentiation and maintenance of I cells, S cells, and basal cells.

**Pparg regulates a diverse set of urothelial genes**. To determine the transcriptional pathways regulated by *Pparg* in the adult urothelium, we performed RNA-sequencing (RNA-Seq) analysis comparing gene expression in urothelial cells isolated from control (*Pparg^fl/fl^*) mice and *Shh^Cre^;Pparg^fl/fl^* mutants during homeostasis (Fig. 4a–c). In addition to *Pparg*, we observed down-regulation of a number of transcription factors, including *Fabp4*, a direct *Pparg* target, *Ppara*, a major regulator of mitochondrial biogenesis, as well as *Grhl3*, a transcription factor that has been shown to regulate S cell differentiation[9] (Fig. 4c).

Gene set enrichment analysis identified up-regulated pathways in *Shh^Cre^;Pparg^fl/fl^* mutants (Fig. 5a). These include innate immune functions ($p = 10^{-9}$), response to Pertussis toxin ($p = 10^{-10}$), virus infection ($p = 10^{-6}$), Toll/IL-1R-domain-containing

adaptor-inducing interferon-β (TRIF)-mediated signaling ($p = 10^{-5}$), and the complement and coagulation cascade ($p = 10^{-7}$). Interestingly, numerous pathways that mediate mitochondrial functions were under-represented in *Shh^Cre^;Pparg^fl/fl^* mutants compared to controls (Figs. 4a and 5; Supplementary Table 1 contains individual *p* values). Pathways most affected were those related to lipid and amino acid metabolism ($p = 10^{-10}$), β-oxidation ($p = 10^{-8}$), fatty acid metabolism ($p = 10^{-9}$), and pyruvate and propanoate metabolism ($p = 10^{-6}$ and $p = 10^{-7}$, respectively). In particular, pathways important for synthesis of cholesterol and unsaturated fatty acids, which include *Ppara* and *Pparg* ligands, were also down-regulated ($p = 10^{-6}$ and $p = 10^{-8}$, respectively).

Fatty acids are mainly metabolized in the mitochondrial matrix through β-oxidation and the tricarboxylic acid (TCA) cycle. However, the mitochondrial membrane is impermeable to fatty acids and a specialized carnitine carrier system consisting of Cpt1, Slc25a20, and Cpt2 control fatty acid transport[29]. We observed down-regulation of all three genes in *Shh^Cre^;Pparg^fl/fl^* mutants (Fig. 4a, Cpt2 immunostaining is shown in Fig. 6a, c). In addition, 15 genes that encode members of complex 1 NADH ubiquinone oxidoreductase were down-regulated, including Nd4, Nd5, and

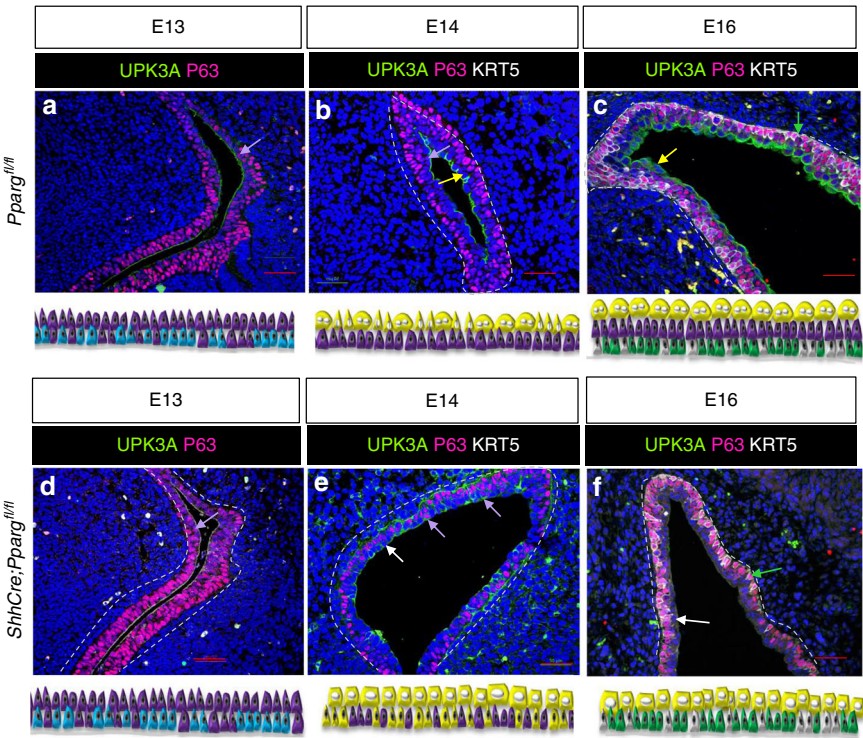

**Fig. 3** *Pparg* controls I cell and S cell development. **a**, **d** Expression of Upk3a and P63 in the urothelium at E13 in a *Pparg^{fl/fl}* control embryo (**a**) and in a *Shh^{Cre};Pparg^{fl/fl}* mutant embryo (**d**). **b**, **e**. Expression of Upk3a, Krt5, and P63 in the urothelium at E14 in a *Pparg^{fl/fl}* control embryo (**b**) and in a *Shh^{Cre}; Pparg^{fl/fl}* mutant embryo (**e**). **c**, **f** Expression of Upk3a, Krt5, and P63 in the urothelium at E16 in a *Pparg^{fl/fl}* control embryo (**c**) and in a *Shh^{Cre};Pparg^{fl/fl}* mutant embryo (**f**). Yellow arrows denote S cells; white arrows denote abnormal S cells; purple arrows denote I cells; and green arrows denote basal cells. Samples used in this experiment: E16 mutant urothelium, *n* = 5; E16 control urothelium, *n* = 4; E14 mutant urothelium, *n* = 7, E14 control urothelium, *n* = 8; E13 mutant urothelium, *n* = 3; E13 control urothelium, *n* = 3. Scale bar: 50 μm

Nd6 that are transcribed from mitochondrial DNA (Fig. 4a). Furthermore, down-regulated were genes encoding members of the complex IV, cytochrome *c* oxidase complex (*Cox1*, *Cox7b*, *Cpx7r*, *Cox16*, and *Cox17*; Fig. 4a, Cox1 immunostaining is shown in Fig. 6b, d; white arrow in Fig. 6d points to an immune cell in the mutant urothelium that is Cox1 positive). Antioxidant proteins that protect against oxidative stress *Sod1* and *Sod2* were also down-regulated in mutants compared to controls (Fig. 4a; immunostaining is shown in Fig. 6e–h).

RNA-Seq analysis reveals down-regulation of genes normally expressed in mature S cells including *Krt20*, *Uchl1*, *Sprr1a*, and *Upks* (Fig. 4c; immunostaining is shown in Fig. 6i–m and Supplementary Fig. 1a–j). We also observed changes in genes important for formation of junctional complexes including *Cldn8* (Fig. 6k, n) and genes involved in vesicle transport (Supplementary Fig. 2a). Electron microscopy (EM) of mutants and controls reveals an abnormal membrane in S cells of *Shh^{Cre};Pparg^{fl/fl}* mutants that lacks the characteristic plaques that line the apical membrane in controls, and vesicles that normally transport Upks to and from the apical membrane were small and abnormally shaped in mutants compared to controls (Supplementary Fig. 2b, c: green-white arrowheads point to morphologically normal fusiform vesicles in controls; pink-white arrowheads point to abnormal vesicles in mutants; the red arrow in Supplementary Fig. 2c points to an abnormal junctional complex in the mutant).

Consistent with the observed expansion of the K14-basal cell population in *Shh^{Cre};Pparg^{fl/fl}* mutants, RNA-Seq analysis revealed up-regulation of *Krt14*, as well as a number of genes expressed in squamous epithelia, including *Krt10*, *Krt13*, *Krt6a*, and *Krt6b* (Fig. 4b). Immunostaining (Fig. 6o, q) shows expression of Krt6a, which marks the basal layers in controls, was present throughout the urothelium in mutants; and *Krt10*, a

marker of cornified epithelia not detectable in the healthy urothelium, was expressed in cells scattered throughout the mutant urothelium (Fig. 6p, r). These findings suggest that *Pparg* is essential in the urothelium for transcriptional control of mitochondrial biogenesis and fatty acid transport, as well as for maintaining proper differentiation of the basal cell, I, and S cell populations.

**Pparg regulates mitochondrial functions in S cells.** Analysis of *Shh^{Cre};Pparg^{fl/fl}* mutants in which *Pparg* is deleted throughout the urothelium revealed abnormalities affecting S cells, I cells, and basal cells. An interesting question is whether *Pparg* plays distinct roles in different epithelial compartments. To begin to address this, we used the tamoxifen-inducible *Upk2CreERT2* line[30] to selectively delete *Pparg* in S and I cells in adults, and then we analyzed the effects on urothelial homeostasis. Analysis of *Upk2CreERT2;Pparg^{fl/fl}* mutants by immunostaining 7 days after tamoxifen induction revealed down-regulation of *Pparg* and *Fabp4*, a direct *Pparg* transcriptional target, indicating that both *Pparg* expression and signaling were decreased in mutants compared to controls (Fig. 7a, b, e, f). This analysis also revealed that Upk1a, Upk3a, which are highly enriched in S cells, and Krt20, which labels mature S cells, were both down-regulated (Fig. 7a–h; Supplementary Fig. 3a–f), suggesting that *Pparg*-dependent transcription regulates S cell differentiation in a cell-autonomous manner. Interestingly, we observed Ki67 expression in 15% of S cells and 10% of I cells in *Upk2CreERT2;Pparg^{fl/fl}* mutants during homeostasis (Fig. 7c, g, i). These findings are surprising, since the adult urothelium is largely quiescent during homeostasis. We also observed exfoliated S cells in urine collected from mutants after tamoxifen induction, but not in controls, suggesting that S

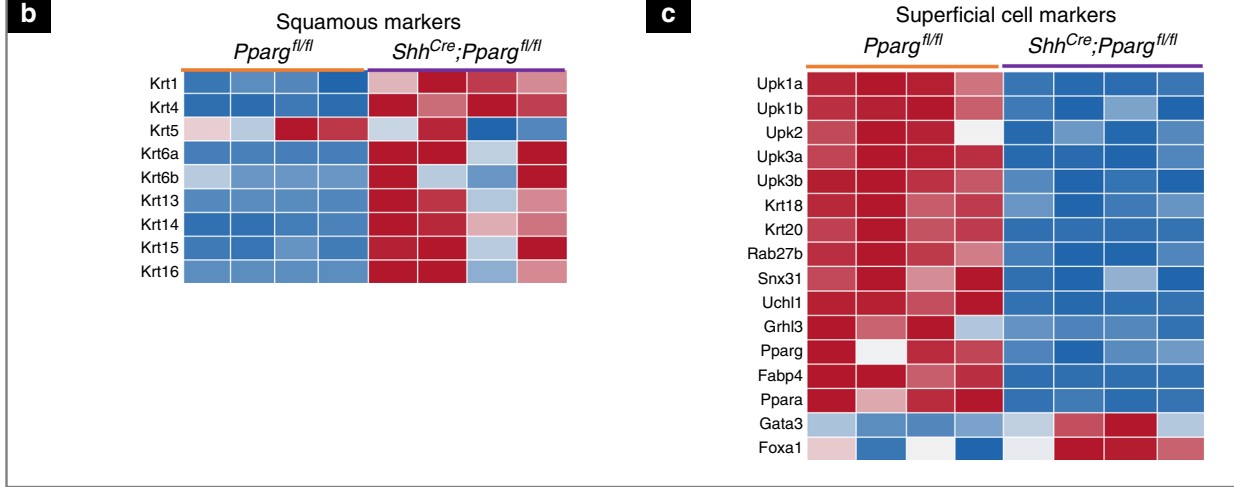

**Fig. 4** Gene expression changes in the urothelium of *Pparg* mutants. **a** A heatmap showing changes in expression of genes related to mitochondrial functions in *Shh^Cre;Pparg^fl/fl* mutants compared to controls. **b** A heatmap showing squamous markers that are up-regulated in *Shh^Cre;Pparg^fl/fl* mutants. **c** A heatmap showing changes in expression of S cell markers in *Shh^Cre;Pparg^fl/fl* mutants compared to controls. *P* values for gene expression changes are listed in Supplementary Table 1. RNA-Seq analysis was performed on RNA isolated from four *Shh^Cre;Pparg^fl/fl* mutants and four *Pparg^fl/fl* controls

cells in the mutants were dying off and being replaced by Ki67-positive I cells and S cells (Fig. 7j, l, black arrowheads in Fig. 7l point to a cluster of S cells). Exfoliated S cells were surrounded by neutrophils (Fig. 7l, black arrow), suggesting that they are in the process of being cleared by the immune system.

To follow the fate of mutant S cells, we pre-labeled S cells of mutants and controls with wheat germ agglutinin (WGA), which binds to the S cell membrane[31]. Animals were then exposed to tamoxifen to induce Cre-dependent recombination, and urine was analyzed each day to determine whether WGA-positive

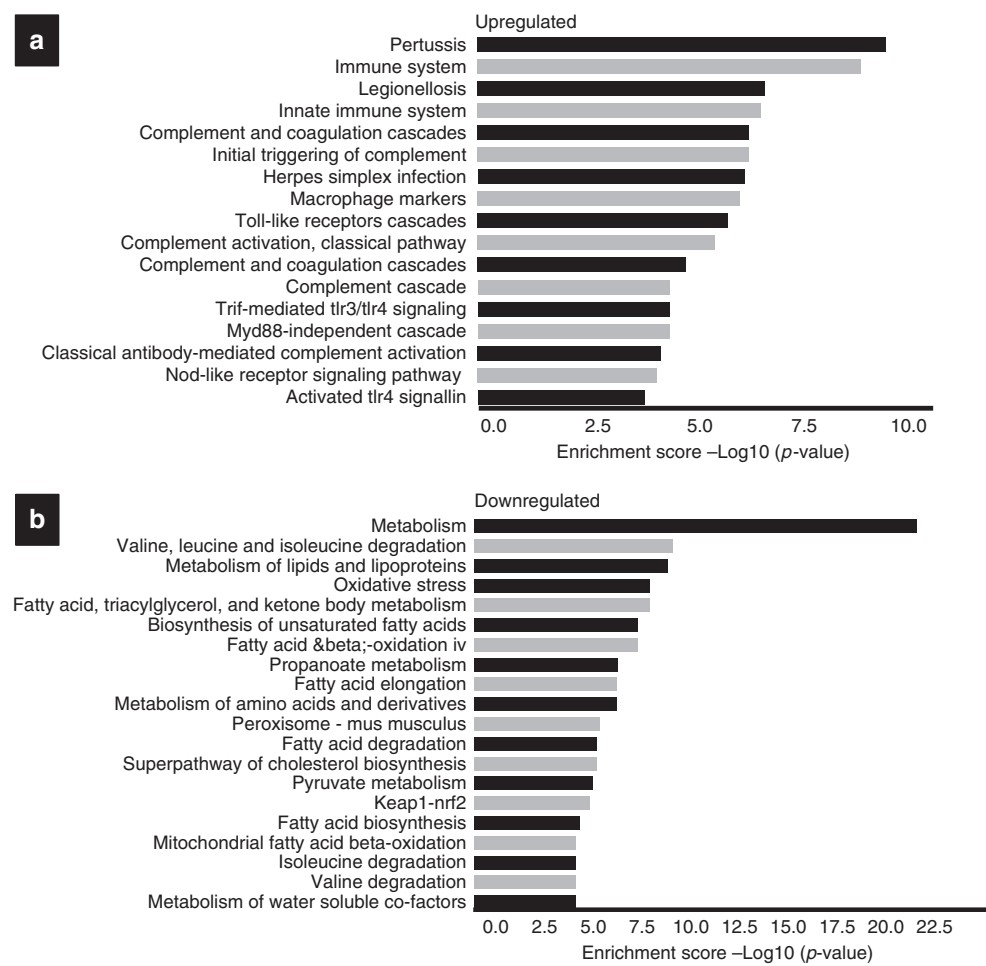

**Fig. 5** Pathway enrichment analysis of $Shh^{Cre};Pparg^{fl/fl}$ mutants compared to controls. **a** Over-representation analysis showing pathways that are up-regulated in the urothelium of mutants vs. controls. **b** Over-representation analysis showing down-regulated pathways in the urothelium of $Shh^{Cre};Pparg^{fl/fl}$ mutants compared to control $Pparg^{fl/fl}$ mice

S cells were present. This analysis revealed few if any WGA-stained (green fluorescent-positive (GFP+)) S cells in urine from control mice; however, urine collected from $Upk2CreERT2;$ $Pparg^{fl/fl}$ mutants contained large numbers of WGA+ S cells (Fig. 7k, m), suggesting that $Pparg$ function during homeostasis is essential for the survival of S cells.

Based on the large number of mitochondrial pathways disrupted in $Shh^{Cre};Pparg^{fl/fl}$ mutants including the complement cascade (Fig. 5a), it would not be surprising if defective S cell differentiation and lethality were linked to mitochondrial defects. To begin to address this, we analyzed ultracellular structure by EM, comparing $Up2CreERT2;Pparg^{fl/fl}$ mutants and controls. Analysis of urothelium in control ($Pparg^{fl/fl}$ mice) revealed normal mitochondria with regularly formed cristae surrounded by the characteristic double membrane (Fig. 7n, o, p; green-white arrowheads point to the double mitochondrial membrane). However, mitochondria in S cells of $Up2CreERT2;Pparg^{fl/fl}$ mutants contained electron-dense inclusions that varied in size (Fig. 7s, t; red-white arrowheads point to inclusions). Higher magnification revealed that the inclusions were located in the intermembrane space in mutants (Fig. 7u, red-white arrowheads designate the position of the inclusion body), suggesting that these are composed of materials that accumulate and fail to enter the mitochondrial matrix, where lipid oxidation takes place.

$Cpt1$, $Cpt2$, and $Sl25a20$ are carnitine palmitoyl transferases that shuttle fatty acids across the mitochondrial membrane to the matrix to initiate β-oxidation[32]. All three genes are down-

regulated in $Pparg$ mutants, suggesting that the inclusions may be composed of fatty acids that accumulate due to impaired mitochondrial transport. To directly address this question, we stained urothelial cells from $Upk2CreERT2;Pparg^{fl/fl}$ mutants and controls with Bodipy, which marks neutral lipids, along with Mitotracker and TOM20, a component of the mitochondrial outer membrane. Bodipy staining, which was undetectable in controls, was clearly visible in mutant mitochondria, labeling large spherical structures (Fig. 7q, r, v, w), indicating that the cargo in mutant inclusions is neutral lipid. These neutral lipids are likely to be fatty acids that are transported across the mitochondrial outer membrane, but fail to reach the mitochondrial matrix, a defect that would severely affect energy metabolism and biomass synthesis.

**$Pparg$ prevents squamous differentiation.** In $Shh^{Cre};Pparg^{fl/fl}$ mutants, the K5-basal population decreases, the K14-basal population expands and squamous markers are up-regulated. EM reveals that the basal population in $Shh^{Cre};Pparg^{fl/fl}$ mutants also displays severe mitochondrial abnormalities, including mitochondrial inclusions (Fig. 8a–d), similar to those observed in S cells. These defects may be secondary to inflammation, loss of barrier function, or could reflect a direct role of $Pparg$ as a regulator of basal cell differentiation. To address this question, we used the tamoxifen-inducible $Tg(Krt5-Cre/ERT2)2Ipc/J$ [ref. [33]; hereafter referred to as $Krt5^{CreER}T2$ mice] to selectively inactivate

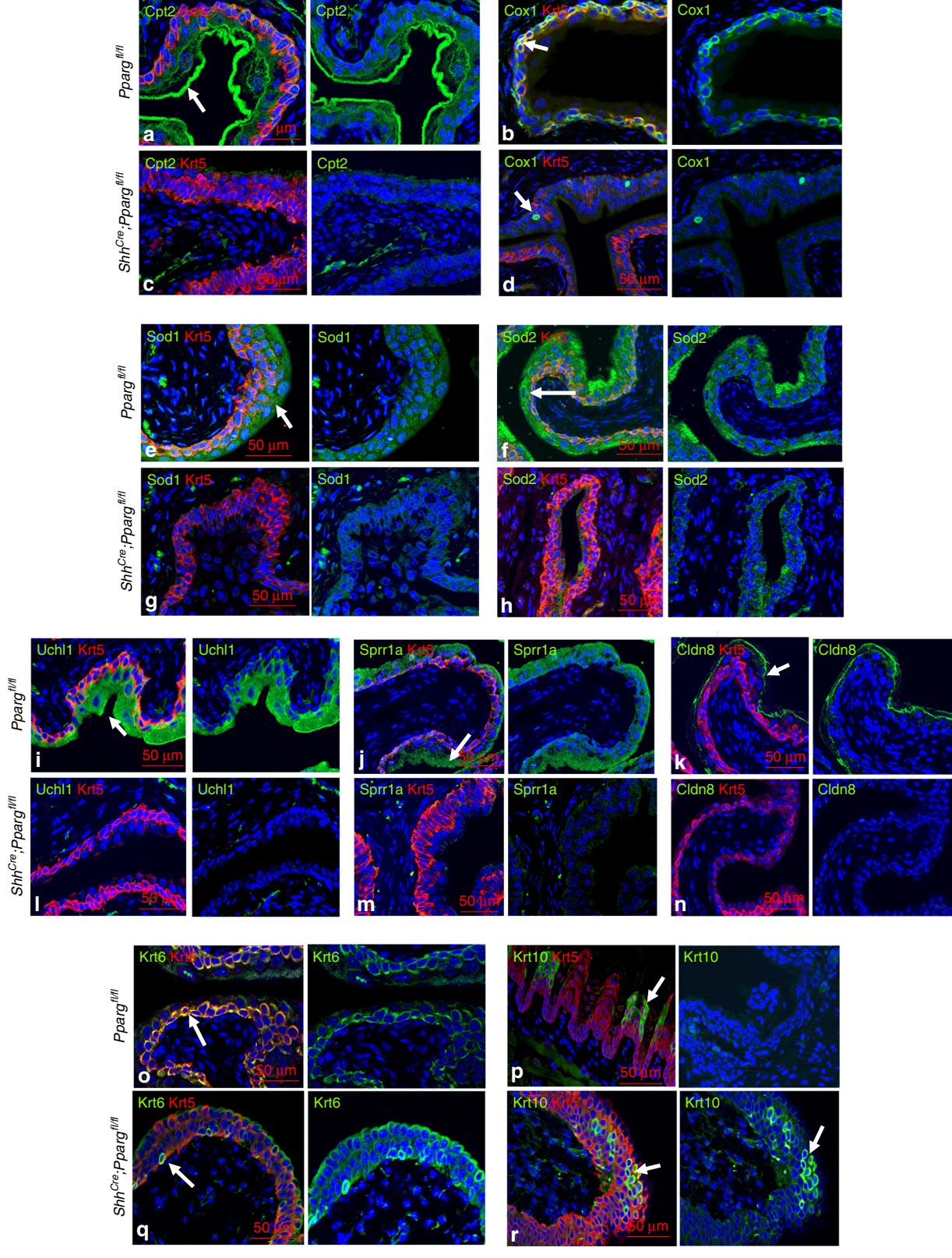

*Pparg* in basal cells and their daughters. Analysis of adult *Krt5*^CreER T2;*Pparg*^fl/fl mutants 14 days after tamoxifen induction revealed down-regulation of *Pparg* in basal cells, while the expression level remained the same in S cells and I cells (Fig. 8e, h). Immunostaining revealed basal cell abnormalities in *Krt5*^CreER T2;*Pparg*^fl/fl mutants similar to those observed in

*Shh*^Cre;*Pparg*^fl/fl mice, including an expanded K14-basal population and up-regulation of Krt10 (Fig. 8f, g, h, i). Consistent with these observations, RNA-Seq analysis of urothelium from mutants and controls revealed up-regulation of squamous markers (Fig. 8k), and over-representation analysis revealed alterations in many of the same pathways as those observed in *Shh*^Cre;

**Fig. 6** Validation of gene expression changes from RNA-Seq experiments. **a**, **c** Expression of Cpt2 and Krt5 in the urothelium of *Pparg^fl/fl* controls (**a**) and *Shh^Cre;Pparg^fl/fl* mutants (**c**). **b**, **d** Cox1 and Krt5 expression in a control (**b**) and in the *Shh^Cre;Pparg^fl/fl* mutant urothelium (**d**). **e**, **g** Sod1 and Krt5 expression in a *Pparg^fl/fl* control urothelium (**e**) and in the urothelium of an *Shh^Cre;Pparg^fl/fl* mutant (**g**). **f**, **h** Sod2 and Krt5 expression in the urothelium of a *Pparg^fl/fl* control (**f**) and in a *Shh^Cre;Pparg^fl/fl* mutant urothelium (**h**). **i**, **l** Uchl1 and Krt5 expression in the urothelium of a wild-type adult tongue (**i**) and in the urothelium of a *Shh^Cre;Pparg^fl/fl* mutant (**l**). **j**, **m** Expression of Sprr1a and Krt5 in the urothelium of a control (**j**) and in a urothelium of a *Shh^Cre;Pparg^fl/fl* mutant (**m**). **k**, **n** Cldn8 and Krt5 expression in a control urothelium (**k**) and in a *Shh^Cre;Pparg^fl/fl* mutant (**n**). **o**, **q** Krt6 and Krt5 expression in a control urothelium (**o**) and in a *Shh^Cre;Pparg^fl/fl* mutant (**q**). **p**, **r** Uchl1 and Krt5 expression in a control urothelium (**p**) and in a *Shh^Cre;Pparg^fl/fl* mutant (**r**) urothelium. Scale bars: 50 μm. Adult wild type, n = 6; adult mutant, n = 5

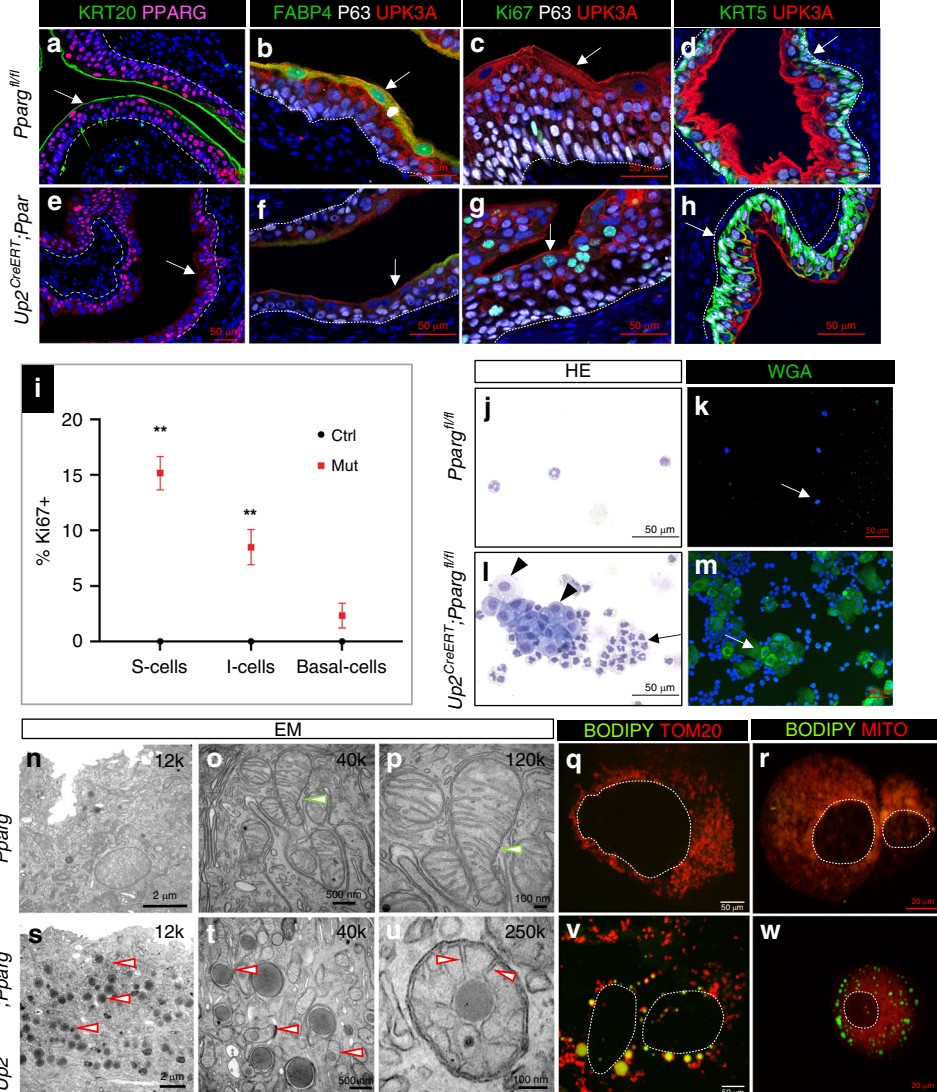

**Fig. 7** *Pparg* is required for S cell differentiation. **a–h** Analysis of the urothelium from adult *Pparg^fl/fl* control mice and *Up2CreERT;Pparg^fl/fl* mutant mice. **a**, **e** Krt20 and Pparg expression. **b**, **f** Fabp4, P63, and Upk3a expression. **c**, **g** Ki67, P63, and Upk3a expression. **d**, **h** Krt5 and Upk3a expression. Dotted lines represent the basement membrane. White arrows point to S cells. **i** Quantification of Ki67+ cells in the urothelium of adult *Pparg^fl/fl* control and *Up2CreERT;Pparg^fl/fl* mutants. A minimum of three independent experiments were performed, and numbers are means of percentages ± SEM. Significance calculated by a two-tailed Student's *t* test; **p < 0.01. **j**, **l** H/E-stained urothelial cells collected from the urine of (**j**) *Pparg^fl/fl* controls and (**l**) *Up2CreERT;Pparg^fl/fl* mutants. Urothelial cells collected from the urine of *Pparg^fl/fl* control (**k**) or *Up2CreERT;Pparg^fl/fl* mutants (**m**) after prestaining with WGA-A488. Scale bars: **a–m**: 50 μm. Transmission electron microscopy (2 μm) of S cells from *Pparg^fl/fl* controls (**n**) and a *Up2CreERT;Pparg^fl/fl* mutant (**s**). Images (500 μm) of mitochondria in S cells of a *Pparg^fl/fl* control (**o**) and *Up2CreERT;Pparg^fl/fl* mutant (**t**). Images (100 nm) of mitochondria in S cells of controls (**p**) and a *Up2CreERT;Pparg^fl/fl* mutant (**u**). Green-white arrowheads in (**o**, **p**) denote mitochondria. Red-white arrowheads in (**s**, **t**) denote mitochondria with inclusions. Red-white arrowheads in **u** denote the position of the inclusion body, which is in the intermembrane space. **q**, **v** Urothelial cells from adult *Pparg^fl/fl* controls and *Up2CreERT;Pparg^fl/fl* mutants stained with BODIPY and TOM20. **r**, **w** Cytospin prep of urothelial cells from an adult *Pparg^fl/fl* control and a *Up2CreERT;Pparg^fl/fl* stained with BODIPY and Mitotracker. Dotted line denotes the nucleus in **q**, **r**, **v**, and **w**. Samples in the analysis: Adult urothelium isolated during homeostasis: *Up2CreERT;Pparg^fl/fl* mutants, n = 8; *Pparg^fl/fl* controls, n = 7. WGA staining and EM were each performed with n = 3 samples from mutants and controls. Scale bar: **q**, **r** 5 μm, **v**, **w** 20 μm. Source data are provided as a Source Data file

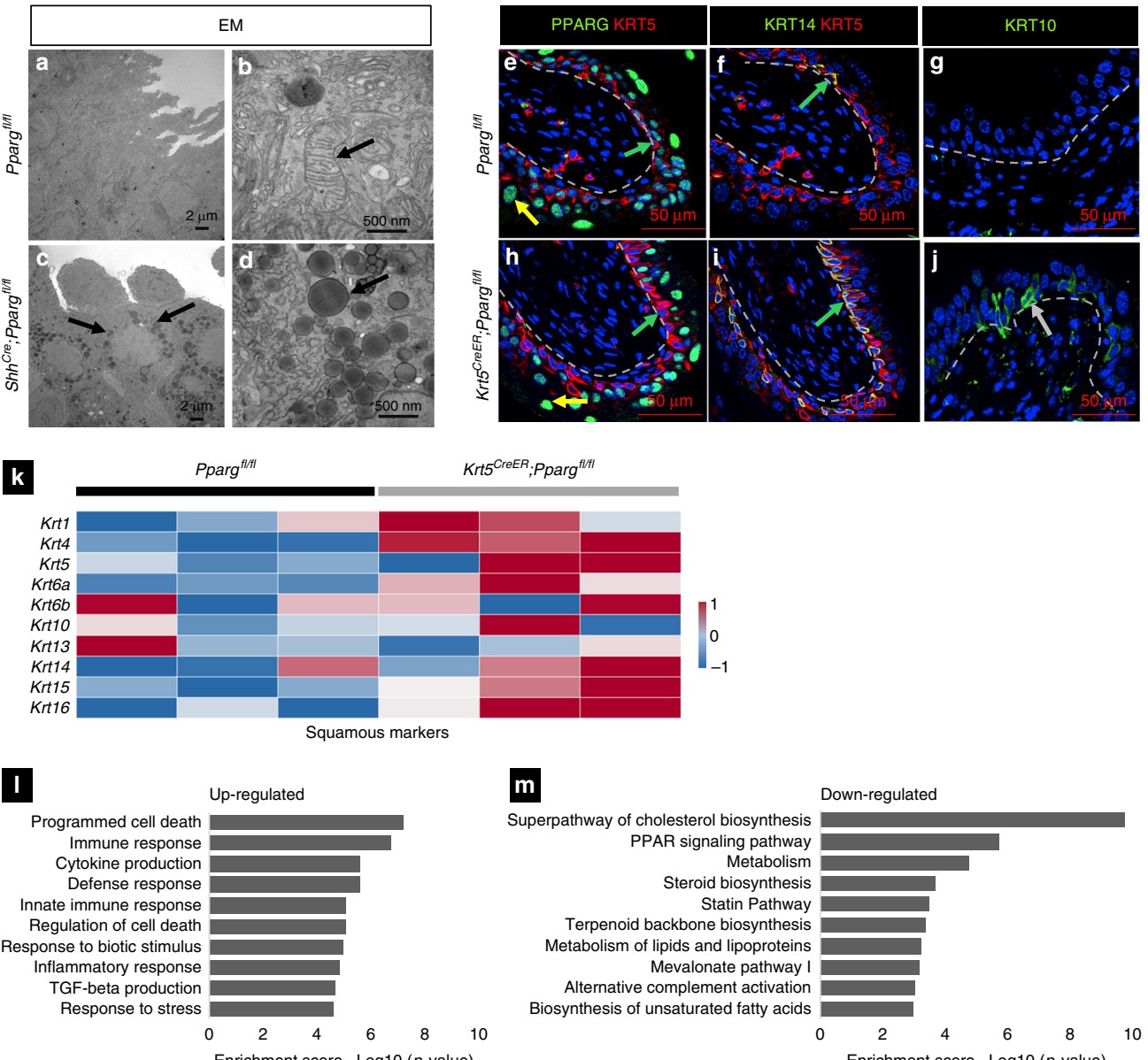

**Fig. 8** Pparg ablation in basal cells results in squamous differentiation. **a–d** Transmission electron microscopy of the urothelium from adult *Pparg*$^{fl/fl}$ controls (**a**, **b**) and *Krt5*$^{CreER}$;*Pparg*$^{fl/fl}$ mutants (**c**, **d**). **a**, **c** Low magnification images showing inclusions detected in basal cells in the urothelium of **c** *Shh*$^{Cre}$; *Pparg*$^{fl/fl}$ mutants but not **a** adult *Pparg*$^{fl/fl}$ controls. **b**, **d** Higher magnification images focusing on the inclusions in basal cells in the urothelium of **d** *Shh*$^{Cre}$; *Pparg*$^{fl/fl}$ that are absent in **b** adult *Pparg*$^{fl/fl}$ controls. This figure shows mitobodies in basal cells that have similar features as those observed in S cells. Scale bar: **a**, **c** 2 μm; **b**, **d** 500 nm. **e–j** Expression of squamous markers in the *Pparg*$^{fl/fl}$ control mice and *Krt5*$^{CreER}$;*Pparg*$^{fl/fl}$ mutant mice 14 days after tamoxifen induction. Pparg and Krt5 expression in a control (**e**) and in a *Krt5*$^{CreER}$;*Pparg*$^{fl/fl}$ mutant (**h**). Krt14 and Krt5 expression in a *Pparg*$^{fl/fl}$ control (**f**) and in a *Krt5*$^{CreER}$;*Pparg*$^{fl/fl}$ mutant (**i**). Krt10 expression in a control (**g**) urothelium and in the urothelium of a *Krt5*$^{CreER}$;*Pparg*$^{fl/fl}$ mutant (**j**). Scale bars: **e–j** = 50 μm. **k** Heatmap showing the changes in genes related to squamous differentiation in basal cells from *Krt5*$^{CreER}$;*Pparg*$^{fl/fl}$ urothelium compared to the *Pparg*$^{fl/fl}$ control urothelium 14 days after tamoxifen induction. **l–m** Pathway enrichment analysis of genes **l** up-regulated and **m** down-regulated in basal cells from adult *Pparg*$^{fl/fl}$ controls compared to *Krt5*$^{CreER}$;*Pparg*$^{fl/fl}$ mutants 14 days after tamoxifen induction. Numbers of samples used for EM; $n = 3$ adult *Shh*$^{Cre}$;*Pparg*$^{fl/fl}$ mutants and $n = 3$ *Pparg*$^{fl/fl}$ controls. Analysis of *Krt5*$^{CreER}$;*Pparg*$^{fl/fl}$ mutants was performed with three mutants and three controls. RNA-Seq analysis of *Krt5*$^{CreER}$;*Pparg*$^{fl/fl}$ was performed with samples from three mutants and three controls. Source data are provided as a Source Data file

*Pparg*$^{fl/fl}$ mutants (Fig. 8l, m). Down-regulated pathways include cholesterol biosynthesis ($p = 10^{-9}$), metabolism ($p = 10^{-5}$), lipid metabolism ($p = 10^{-3}$), fatty acid metabolism and biosynthesis ($p = 10^{-3}$), and *Pparg* signaling ($p = 10^{-6}$). Up-regulated pathways include programmed cell death ($p = 10^{-7}$), immune response ($p = 10^{-6}$), cytokine production ($p = 10^{-5}$), and stress response ($p = 10^{-4}$). Taken together, these observations suggest that Pparg is important in basal cells for regulating differentiation, as well as for controlling immune and metabolic pathways.

**Pparg controls NF-κB signaling in the urothelium**. During homeostasis, about 90% of basal cells are K5-basal cells, while K14-basal cells make up the remaining 10% of the population in the wild-type urothelium. We observed expansion of the K14-basal population in adult *Shh*$^{Cre}$;*Pparg*$^{fl/fl}$ mutants during homeostasis that persists for as long as 1 year; however, we did not observe evidence of squamous metaplasia or tumor formation during homeostasis (Supplementary Fig. 1k–n). The urothelium is largely quiescent during homeostasis, but can rapidly

regenerate in response to toxins or bacterial infection. We therefore investigated whether phenotypes observed in mutants during homeostasis would be exacerbated during regeneration, using a mouse model of UTI. Animals were infected by intra-urethral catheterization with UTI 89, a UPEC isolated from a patient with cystitis[34]. UPEC infects and multiplies within S cells, inducing cell death and exfoliation (Supplementary Fig. 4). During normal regeneration in wild-type controls, we observed activation of NF-κB signaling, evidenced by nuclear expression of p65/Rela, which is detected within 3 h post infection (Supplementary Fig. 4a). Proliferation based on expression of Ki67 reaches highest levels in basal cells and I cells 24 h post infection (Supplementary Fig. 4c, d). Newly formed S cells are observed at 72 h post infection (Supplementary Fig. 4e), a stage when proliferation decreases to near-homeostatic levels in the I cell and B cell compartments. Mature Krt20+ S cells are observed about 2 weeks after infection (Supplementary Fig. 4f).

Analysis of wild-type mice 24 h post infection revealed extensive proliferation throughout the basal and I cell compartments (Fig. 9a) and the number of Krt14-expressing cells dramatically increased compared to uninfected animals (Fig. 9b, c). Analysis of $Shh^{Cre}$; $Pparg^{fl/fl}$ mutants 24 h post infection revealed extensive proliferation (Fig. 9e) and the kinetics of bacterial infection were similar in mutants compared to controls, indicating that the $Shh^{Cre}$;$Pparg^{fl/fl}$ mutant urothelium responds to infection (Fig. 9i). The Krt14-expressing basal cell population, which was increased during homeostasis in $Shh^{Cre}$;$Pparg^{fl/fl}$ mutants compared to controls (Fig. 9f; Supplementary Fig. 5a–h), was further expanded after UTI, occupying all layers of the mutant urothelium (Fig. 9g; Supplementary Fig. 5i–v). Intriguingly, in controls, the number of K14-basal cells transiently increased at 24 h and decreased to homeostatic levels by 72 h, but in mutants, the expanded population persisted long for months, long after infection was cleared (Fig. 9d, h; Supplementary Fig. 5i–v). Analysis of $Shh^{Cre}$; $Pparg^{fl/fl}$ mutants 6 weeks after infection revealed severe edema (Fig. 9r), which was not present in controls (Fig. 9j). We also observed up-regulated expression of squamous markers Krt14, Krt5, and Krt10 by immunostaining (Fig. 9k, l, m, s, t, u) results also observed in RNA-Seq analysis (Fig. 9z). In addition, we detected increased expression of Ki67 in mutants, which was not observed in $Pparg^{fl/fl}$ controls (Fig. 9n, v).

Interestingly, analysis 6 weeks post infection revealed alterations in the mutant urothelium that are associated with invasion. Slug, which labels cells undergoing epithelial–mesenchymal transition, was up-regulated in the mutants compared to controls (Fig. 9o, w), and laminin staining, which revealed an intact basement membrane in controls, was patchy or absent in mutants, suggesting that the basement membrane was compromised (Fig. 9p, x). We also observed up-regulation of Smooth muscle α-actin, a marker of fibrosis, in mutants, which is low or barely detectable in controls (Fig. 9q, y).The persistent inflammation and abnormal differentiation of the basal population was not due to re-infection, since the animals were treated with antibiotics at 30 h post infection, and the bacterial load followed a similar pattern of kinetics in mutants and controls, where colony-forming unit (CFU/ml) was highest 24 h post infection, and low by 4 weeks post infection (Fig. 9i).

To begin to elucidate how $Pparg$ loss-of-function leads to squamous-like differentiation and other changes in the mutant urothelium, we performed RNA-Seq analysis of urothelium isolated from $Shh^{Cre}$;$Pparg^{fl/fl}$ mutants and controls 4 weeks after infection. These analyses revealed up-regulation of a number of genes in the NF-κB signaling pathway compared to controls (Fig. 10a). To confirm that NF-κB signaling was aberrantly induced in mutants, we performed immunostaining with p65/Rela. Immunostaining analysis reveals nuclear p65/Rela

expression both in mutants and controls at 24 h, as expected (Fig. 10a, e). In controls, NF-κB signaling was down-regulated by 72 h based on the absence of nuclear p65/Rela expression (Fig. 10c, d). In $Shh^{Cre}$;$Pparg^{fl/fl}$ mutants, however, nuclear p65/Rela expression persisted throughout the urothelium for months post infection (Fig. 10f, g), which would be likely to result in persistent inflammation. Consistent with this, we observed massive edema in $Shh^{Cre}$;$Pparg^{fl/fl}$ mutants weeks after UTI (Fig. 10h, m), and extensive infiltration of leukocytes revealed by CD45 staining in the mutant urothelium and stroma, which was not observed in controls. (Fig. 10i, n). Phenotyping the immune cells by macrophage marker F4/80, T cell marker CD3 and B cell marker CD19 revealed that the infiltrating cells are largely composed of macrophage (Fig. 10j, o) and T cells (Fig. 10k, p) in mutants, but few if any B cells. (Fig. 10l q). The persistent inflammation in $Pparg$ mutants was unlikely to reflect a direct role for $Pparg$ in immune cells, since flow cytometry confirmed that the $Shh^{Cre}$ promoter used to delete $Pparg$ was active in urothelial cells, but not in leukocytes (Supplementary Fig. 6). Persistent inflammation can induce squamous or abnormal differentiation in the urothelium, for example, in patients with indwelling catheters[35]. Whether the persistent activation of the NF-κB signaling pathway underlies the abnormal urothelial differentiation in $Shh^{Cre}$;$Pparg^{fl/fl}$ mutants is an interesting possibility.

Taken together, our observations indicate that $Pparg$ plays critical and distinct roles in development, homeostasis and regeneration of the urothelium. $Pparg$ signaling controls S cell differentiation and survival during development and homeostasis, prevents squamous differentiation in the basal compartment and regulates immune responses during regeneration after UTI. We did not observe tumor formation in mutants, but we did observe a number of changes that are associated with the basal subtype of urothelial carcinoma, including squamous differentiation and expression of EMT markers.

## Discussion

In this study, we show that $Pparg$ plays an essential role as a regulator of urothelial development in vivo, controlling differentiation and/or survival of basal cells, I cells and S cells. Our findings suggest that $Pparg$ is an important transcriptional regulator of mitochondrial biogenesis and fatty acid transport in urothelial cells, functions required for β-oxidation. Our studies also suggest that $Pparg$ plays a critical role in suppressing squamous differentiation and resolving the innate immune response in the urothelium after injury.

$Pparg$ is a lipid sensor and is known to regulate energy metabolism in many cell types, including adipocytes, endothelial cells, hepatocytes, and macrophages[36], and has also been suggested to be a regulator of lipid metabolism in basal/SCCL tumors[26]. There are three $Ppar$ family members, $Ppara$, $Pparb/d$, and $Pparg$, and it is generally accepted that these transcription factors have distinct roles in metabolic regulation of liver, muscle, and adipose tissue. A number of studies suggest that $Ppara$ controls fatty acid oxidation and mitochondrial biogenesis, while $Pparg$ regulates the adipogenesis program and controls insulin sensitivity in patients with type 2 diabetes[36,37]. However, $Pparg$ has been shown to regulate a wide range of mitochondrial functions in epithelial cells of the renal proximal tubule and in neurons[38,39]. Our studies suggest that a similar situation may exist in the urothelium, where $Pparg$ regulates expression of genes important for lipid and amino acid metabolism, β-oxidation and fatty acid metabolism. We observe down-regulation of $Cpt1$, $Cpt2$, and $Slc25a20$ in mutants, transporters that shuttle fatty acids into the mitochondria matrix, a rate-limiting step in fatty acid oxidation[29]. The presence of

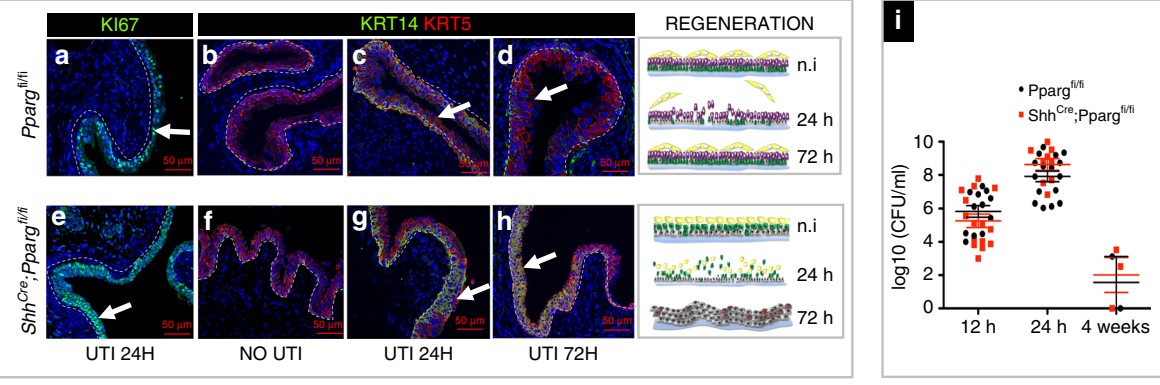

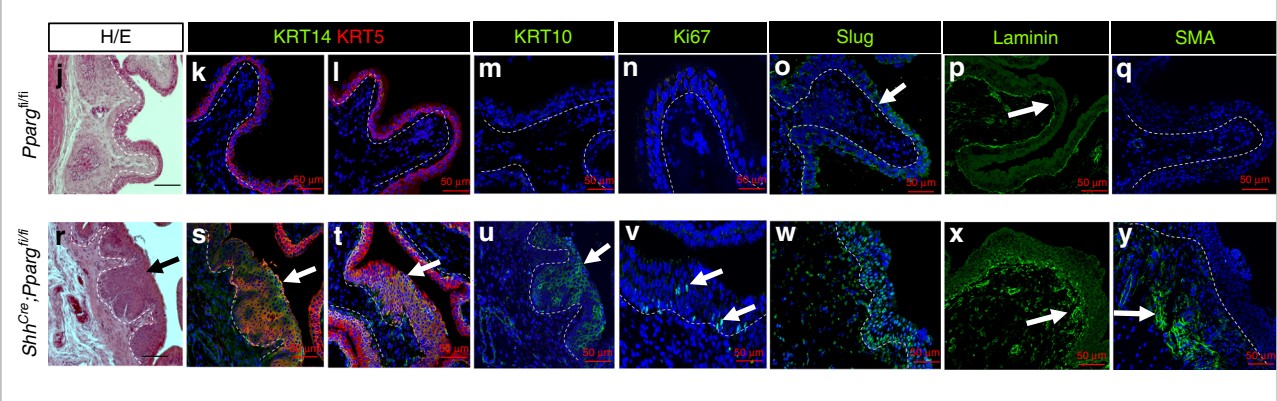

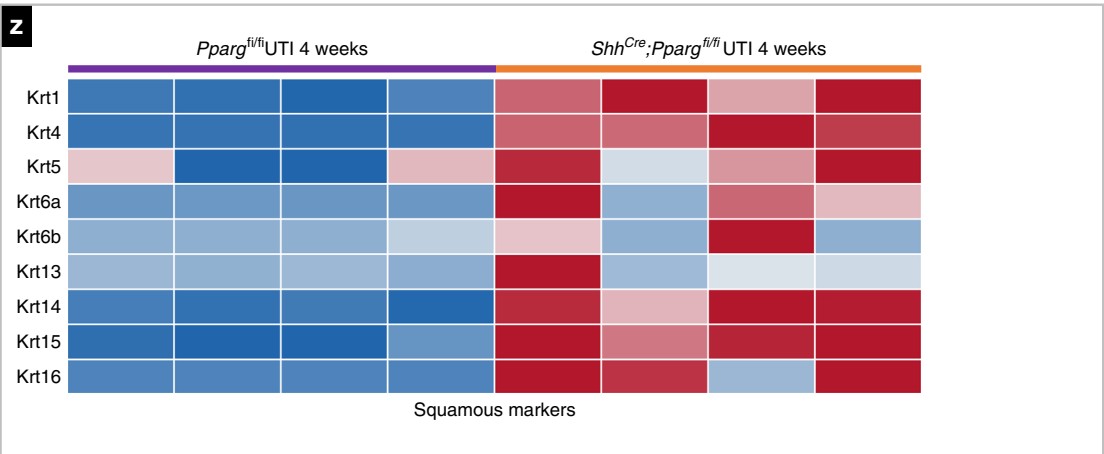

Fig. 9 UTI-induced injury leads to squamous-like differentiation and persistent inflammation in *Pparg* mutants. Expression of Ki67 in the urothelium of a *Pparg*fl/fl control (**a**) and in a *Shh*Cre;*Pparg*fl/fl mutant (**e**) 24 h after UTI. Expression of Krt14 and Krt5 in adult *Ppargfl/fl* control mice and in *Shh*Cre;*Ppargfl/fl* mutant mice. 0 h (**b**, **f**) 24 h (**c**, **g**), and 72 h (**d**, **h**) post infection. Right panels show a schematic representation of UTI-induced regeneration in the urothelium of controls (top) *and Shh*Cre;*Pparg*fl/fl mutants (bottom). (**i**) Bacterial outgrowth (CFU) in the bladder of *Ppargfl/fl* control mice or *Shh*Cre;*Ppargfl/fl* mutant mice at 12 h, 24 h, and 4 weeks post infection. **j**–**y** Analysis of *Ppargfl/fl* control and *Shh*Cre;*Ppargfl/fl* mutants 4 weeks post-UTI. Hematoxylin and eosin staining of a urothelium from an adult control *Pparg*fl/fl mouse (**j**) and a *Shh*Cre;*Ppargfl/fl* mutant mouse (**r**). Expression of Krt14 and Krt15 in the urothelium of an adult control *Pparg*fl/fl mouse (**k**, **l**) and a *Shh*Cre;*Ppargfl/fl* mutant mouse (**s**, **t**). Expression of Krt10 in the urothelium of an adult control *Pparg*fl/fl mouse (**m**) and a *Shh*Cre;*Ppargfl/fl* mutant mouse (**u**). Expression of Ki67 in the urothelium of an adult control *Pparg*fl/fl mouse (**n**) and a *Shh*Cre;*Pparg*fl/fl mutant mouse (**v**). Expression of Slug in the urothelium of an adult control *Pparg*fl/fl mouse (**o**) and a *Shh*Cre;*Ppargfl/fl* mutant mouse (**w**). Expression of laminin in the of a urothelium of an adult control *Pparg*fl/fl mouse (**p**) and a *Shh*Cre;*Pparg*fl/fl mutant mouse (**x**). Expression of SMA in the urothelium of an adult control *Ppargfl/fl* mouse (**q**) and a *Shh*Cre;*Pparg*fl/fl mutant mouse (**y**). (**z**) A heatmap based on RNA-Seq analysis of *Shh*Cre;*Ppargfl/*fl mutants vs. control mice 4 weeks after UTI showing expression of squamous markers. Scale bars: 50 μm. The number of mutants and controls was analyzed: UTI 24 h *Pparg*fl/fl, $n = 3$; *Shh*Cre;*Ppargfl/fl*, $n = 3$; UTI 72 h *Pparg*fl/fl, $n = 4$; *Shh*Cre;*Ppargfl/fl*, $n = 4$; UTI 4 weeks *Pparg*fl/fl, $n = 8$; *Shh*Cre;*Ppargfl/fl*, $n = 11$; RNA-Seq UTI 4 weeks *Pparg*fl/fl, $n = 5$; *Shh*Cre;*Ppargfl/fl*, $n = 5$. Source data are provided as a Source Data file

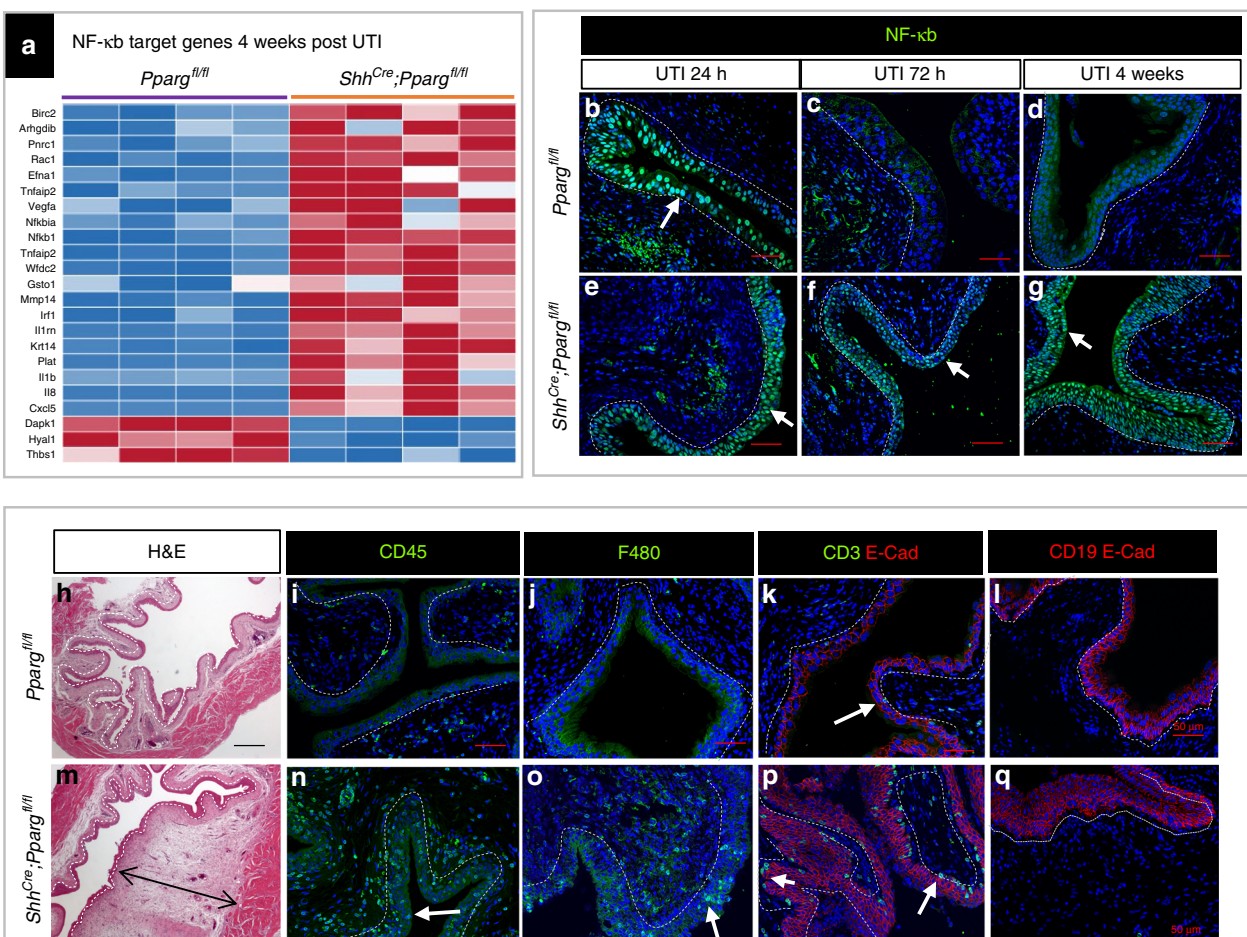

**Fig. 10** Pparg is required for proper regulation of NF-κB. **a** Heatmap showing changes in NF-κB target genes in the urothelium of *Shh^{Cre};Pparg^{fl/fl}* mutants compared to *Pparg^{fl/fl}* controls 4 weeks after UTI. **b–d** Expression of NF-κB (Rela), in the urothelium of adult *Pparg^{fl/fl}* control mice 24 h after UTI (**b**), 72 h after UTI (**c**), and 4 weeks after UTI (**d**). **e–g** Expression of NF-κB (Rela) in the urothelium of a *Shh^{Cre};Pparg^{fl/fl}* mutant 24 h post UTI (**e**), 72 h post UTI (**f**), and 4 weeks post infection (**g**). White arrows pointed to urothelial cells with nuclear Rela staining. Analysis of *Pparg^{fl/fl}* controls (**h–l**) and *Shh^{Cre};Pparg^{fl/fl}* mutants (**m–q**) 4 weeks after UTI. Hematoxylin and eosin staining of the urothelium of an adult *Pparg^{fl/fl}* control mouse (**h**) and a *Shh^{Cre};Pparg^{fl/fl}* mutant mouse (**m**). CD45 expression in the urothelium of an adult *Pparg^{fl/fl}* control mouse (**i**) and a *Shh^{Cre};Pparg^{fl/fl}* mutant mouse (**n**). F480 expression in the urothelium of an adult *Pparg^{fl/fl}* control mouse (**j**) and a *Shh^{Cre};Pparg^{fl/fl}* mutant mouse (**o**). CD3/Ecad expression in the urothelium of an adult *Pparg^{fl/fl}* control mouse (**k**) and a *Shh^{Cre};Pparg^{fl/fl}* mutant mouse (**p**). CD19/Ecad expression in the urothelium of a *Pparg^{fl/fl}* control mouse (**l**) and a *Shh^{Cre};Pparg^{fl/fl}* mutant mouse (**q**). The black arrow in (**m**) denotes edema in the bladder of *Shh^{Cre};Pparg^{fl/fl}* mutants. White arrows in (**b, e, f, g**) point to cells expressing leukocyte markers. Scale bars: 50 μm. Numbers of animals for experiments: n = 3 *Shh^{Cre};Pparg^{fl/fl}* mutants and three controls were analyzed at 24 h post infection, n = 4 *Shh^{Cre};Pparg^{fl/fl}* mutants and n = 4 controls were analyzed at 72 h post infection, n = 8 controls and n = 11 *Shh^{Cre};Pparg^{fl/fl}* mutants were analyzed at 4 weeks post infection. For RNA-Seq experiments, n = 5 *Shh^{Cre};Pparg^{fl/fl}* mutants and n = 5 controls were analyzed

lipid-containing inclusions in the intermembrane space of mitochondria in the mutants suggests that mutant urothelial cells have a diminished ability to carry out fatty acid oxidation, which is likely to reduce energy availability.

Our studies further revealed that *Pparg* is important for survival of S cells, which are progressively shed from the mutant urothelium in adult *Upk2CreERT2;Pparg^{fl/fl}* mutants after tamoxifen induction. S cells normally maintain a complex vesicle transport system that shuttles Upks crystals and apical membrane to endosomes for degradation when the bladder contracts during voiding. When the bladder fills, the length of the apical surface of S cells expands dramatically[3]. This expansion depends on de novo synthesis of Upks, which are assembled into crystals and transported by specialized vesicles from the Golgi to the apical surface[40,41], a process that occurs several times each day. S cells are long-lived, enormous, and polyploid and are likely to require a substantial amount of energy to produce and transport biomass,

and hence may be particularly sensitive to mitochondrial defects. On the other hand, structural alterations and lipid accumulation in the mitochondria may lead to cell death, as has been observed in the aging bladder, where lipofuscin accumulation in S cells is lethal due to a decreased antioxidant capacity[42]. Progressive accumulation of lipid and reactive oxidative species could contribute to various age-related defects in urothelial function as well as cancer and other diseases.

*Pparg* is known to be an important regulator of inflammatory response, in part by regulating transcriptional activity of NF-κB, which among other things, controls the innate immune response to UPEC infection. *p65/Rela*, one of five NF-κB family members, is transiently up-regulated in the wild-type urothelium in response to UPEC infection, but persists in *Shh^{Cre};Pparg^{fl/fl}* mutants for months accompanied by edema and leukocyte infiltration (Fig. 5), suggesting that mutants fail to resolve the NF-κB response. NF-κB regulates expression of immune genes, and also

plays an important role in epithelial barriers such as the skin, gut, and esophagus, controlling recognition and response to invading pathogens[43–45]. NF-κB signaling, as evidenced by p65 expression, is activated rapidly in the wild-type urothelium in response to UPEC infection. Recent studies suggest that its initial activation may be triggered by binding of the bacterial adhesin, Fimh, to Upk1a, which is expressed on the surface of S cells. This inter-action triggers Toll-like receptor 4-mediated pattern recogni-tion[46]. *Pparg* regulates NF-κB signaling by transrepression, either by binding directly to the NF-κB protein, which prevents its interaction with promoter regions of target genes, or alternately, SUMOylated *Pparg* can bind to the nuclear receptor corepressor complex on the promoter region of NF-κB target genes to prevent the dissociation of co-repressors, which is required in gene activation[47,48]. While the direct mechanism by which *Pparg* controls NF-κB in the urothelium is unclear, our studies pro-vide strong evidence that *Pparg* regulates *p65/Rela* expression and is required in urothelial cells to suppress the innate immune response induced by UPEC infection.

Positive and negative *Pparg* signaling can have profound effects on bladder cancer cells[49] and on immune functions in MIBC[50]. *Pparg* expression is down-regulated in the basal subtype of MIBC compared to the healthy urothelium, and up-regulated in the luminal subtype of MIBC. We observed a number of abnormal-ities in *Pparg* mutants that are similar to those observed in MIBC subtypes with low *Pparg* expression, including increased expres-sion of basal/squamous markers (Krt14, Krt6, Krt5), persistent NF-κB signaling[25,51], and up-regulation of pathways activated during invasion (Snail1, Slug, and vimentin) as well as a com-promised basement membrane (Fig. 9). Despite these similarities, we did not observe tumor formation in *Shh^Cre;Pparg^fl/fl* mutants, suggesting that *Pparg* mutations are unlikely to be primary dri-vers of tumor formation. Whether mutations in *Pparg* mutations promote tumorigenesis in cooperation with other mutations, or contribute to dis-regulated differentiation and immune functions after tumor initiation are interesting possibilities.

## Methods
**Mice**. *Shh^Cre* mice (B6.Cg-Shhtm1(EGFP/cre)Cjt/J)[28] and *mTmG^fl/fl* (Gt(ROSA) 26Sortm4(ACTB-tdTomato,-EGFP)Luo/J)[52] mice were obtained from Jackson Laboratory (stock #005622, #007576). *K5CreERT2* mice (FVB.Cg-Tg(KRT5-cre/ERT2)2Ipc/JeldJ)32[33] were obtained from D. Metzger and P. Chambon. *Upk2CreERT2* mice (B6;CBA-Tg(Upk2-icre/ERT2)1Ccc) were generated in the Cordon-Cardo lab[30]. *Pparg^fl/fl* mice[27] were obtained from Dr. Ira Goldberg. All work with mice was approved by and performed under the regulations of the Columbia University Institutional Animal Care and Use Committee. Animals were housed in the animal facility of Irving Cancer Research Center, Columbia University.

**Genotyping**. Genotyping was by PCR analysis of tail DNA. Primers for genotyping *Pparg^fl/fl* were: 5′-CTCCAATGTTCTCAAACTTAC-3′ (forward) and 5′- GAT-GAGTCATGTAAGTTGACC-3′ (reverse), generating a 285 bp product from floxed allele and a 250 bp product from wild-type allele. Primers for genotyping *Shh^Cre* mice were: 5′-TGATGAGGTTCGCAAGAACC-3′ (forward) and 5′-CCATGAGTGAACGAACCTGG-3′ (reverse), generating a 400 bp product. Pri-mers for genotyping *Upk2CreERT2* mice were 5′-GCGGGAGTTCCAGAAAGAG-3′ (common), 5′-AGGACAGCCAGCAGA ATCAG-3′ (wild type), and 5′-AGATCTCCTGTGCAGCATG-3′ (mutant), generating a 250 bp product from wild-type allele and a 290 bp product from floxed allele. Primers for *Krt5^CreERT2* mice were 5′-ATTTGCCTGCATTACCGGTC-3′ (forward) and 5′-ATCAACGTTTTGTTTTCGGA-3′ (reverse), generating a 350 bp product. Primers for genotyping *mTmG* mice were: 5′-CTCTGCTGCCTCCTGGCTTCT-3′ (com-mon), 5′-TCAATGGGCGGGGGTCGTT-3′ (mutant), and 5′-CGAGGCGGAT-CACAAGCAATA-3′ (wild type), generating a 330 bp product from wild-type allele and a 250 bp product from floxed allele.

**Tamoxifen and 4-OHT administration**. Male and female adult *Upk2CreERT2; Pparg^fl/fl* mice (8–12 weeks of age) were injected with tamoxifen (Sigma, cat# T5648), intraperitoneally, at a dose of 5 mg per 30 g body weight three times over a period of 7 days. Male and female *Krt5^CreER and Pparg^fl/fl* mice (8–12 weeks of age)

were injected with tamoxifen, intraperitoneally, at a dose of 5 mg per 30 g body weight for five consecutive days.

**UTI with UPEC**. UPEC strain UTI 89 (a gift from the Hultgren Lab) was isolated from a patient with an acute bladder infection[53]. Adult female mice (8–14 weeks) were anesthetized with isoflurane, and inoculated via transurethral catheterization with 75 μl of bacterial suspension ($10^7$ CFU/ml) in phosphate-buffered saline (PBS) or 75 μl sterile PBS according. Urine was collected 12 and 24 h post infection, assayed for bacterial counts, and analyzed using the Cytospin and Hema3 staining Kit (Fisher Scientific). Titers of $10^6$–$10^7$ CFU/ml were considered to be a robust infection. Sulfatrim (240 mg/kg) was administered 30 h after inoculation with UTI 89 to avoid re-infection. At the indicated times, mice were sacrificed, their bladders were aseptically removed, and processed for microscopy and histology.

**Analysis of Cre-dependent recombination in leukocytes**. *Shh^Cre;Pparg^fl/fl*; *mTmG* mice were generated by intercrossing *Shh^Cre;Pparg^fl/fl* mice with *mTmG^fl/fl* (Gt(ROSA)26Sortm4(ACTB-tdTomato,-EGFP)Luo/J)[52] mice. In this line, cells that undergo Cre-dependent recombination will express membrane-bound Gfp, and cells that do not undergo recombination will express membrane-bound Tomato. *Leukocytes*: To obtain leukocytes, samples were collected from *Shh^Cre;Pparg^fl/fl* ; mTmG mice via cardiac puncture and red blood cells were lysed in ACK lysis buffer at room temperature for 5 min. Immune cells were then collected by cen-trifugation at $300 \times g$ for 5 min in an Eppendorf centrifuge. Pellets were re-suspended in 300 μl FACS buffer and then passed through a 35 μm filter. *Urothelial cells*: Bladders were dissected into OPTI-MEM media, opened and transferred to a solution of 20 mM EDTA solution in PBS, and incubated for 20 min to loosen the urothelium from the stroma. Bladders were then transferred to fresh OPTI-MEM media and the urothelium were manually removed from the stroma. Medium containing urothelial cells was then centrifuged at $500 \times g$ for 5 min at 4 °C, in an Eppendorf centrifuge. The supernatant was discarded, and the pellets were re-suspended in 500 μl of 0.25% Trypsin-EDTA (Thermo Fisher, #25200056) and incubated on a heating block at 37 °C for 25 min with trituration every 5 min. Trypsin was neutralized by adding 500 μl Dulbecco's modified Eagle's medium: nutrient mixture F-12 to the cell suspension. Urothelial cells were collected by centrifugation at $500 \times g$ for 5 min at 4 °C. Supernatants were discarded and pellets were re-suspended in 300 μl FACS buffer and then passed through a 35 μl filter. Single-cell suspension was obtained by resuspending pellets in 300 μl FACS buffer, after which the suspension was passed through a 35 μl filter. Cells were analyzed by a BD Aria II Cell sorter using 30 psi pressure and 100 μm nozzle aperture.

**RNA-sequencing**. For *Shh^Cre;Pparg^fl/fl* mice, bladders were dissected into OPTI-MEM media, and then transferred to a solution of 20 mM EDTA solution in PBS and incubated for 20 min Bladders were then transferred to fresh OPTI-MEM media and the urothelium was manually separated from the stroma. The media containing urothelial cells were centrifuged at $500 \times g$ for 5 min at 4 °C in an Eppendorf 5417C Centrifuge. The supernatant was discarded, and the pellet was processed for total RNA extraction. For *Krt5^CreER;mTmG;Pparg^fl/fl* mice, pellets were re-suspended in 300 μl FACS buffer to produce a single-cell suspension. The cell suspension was filtered through a 35 μm filter and then sorted on a BD Aria II Cell cell sorter using 30 psi pressure and a 100 μm nozzle aperture to collect GFP-positive cells. Cells were then centrifuged at $500 \times g$ for 10 min at 4 °C. The supernatant was discarded, and the pellet was processed for total RNA extraction. Samples containing 100 ng and a RIN (regulation identification number) >8 were used for RNA-Seq. Messenger MRNA were enriched using poly-A pulldown before proceeding to library preparation using Illumina TruSeq RNA prep kit. Libraries were then sequenced using Illumina HiSeq2500/HiSeq4000 at the Columbia Genome Center. Thirty million single-end 100 bp reads were acquired per sample.

Sequencing data was processed by RTA (Illumina) for base calling and bcl2fastq2 (version 2.17) for converting BCL to fastq format, coupled with adaptor trimming. Then, the reads were mapped to mouse: UCSC/mm10 as the reference genome using STAR (2.5.2b) and feature Counts (v1.5.0-p3). Differentially expressed genes were identified using DESeq, an R package based on a negative binomial distribution that models the number reads from RNA-Seq experiments and tests for differential expression. Differentially expressed genes were filtered by average expression level (fragments per kilobase of transcript per million mapped reads) >10, differential expression >2-fold, and adjusted $p$ value <0.05 by Benjamini–Hochberg multiple testing correction. Gene ontology categories were obtained with $q$ values <0.05 by Benjamini–Hochberg multiple testing correction. Over-representation analysis was performed on gene sets from RNA-Seq data obtained from analysis of controls vs. *Shh^Cre;Pparg^fl/fl* mice, *Upk2CreERT2;Pparg^fl/fl* mice, and *Krt5^CreER;Pparg^fl/fl* mice, respectively. We used gene set analysis with the ConsensusPathDB (http://cpdb.molgen.mpg.de/MCPDB), $p$ values were set at 0.01 for all over-representation analysis analyses.

**Immunostaining**. Bladders were embedded in paraffin and serial sections were generated. For immunohistochemistry, paraffin sections were deparaffinized using HistoClear and rehydrated through a series of ethanol and 1× PBS washes. Antigen retrieval was performed by boiling slides for 15 min in pH 9 buffer or 30 min in pH 6 buffer. Primary antibodies in 1% horse serum were incubated overnight at 4 °C.

The next day, slides were washed with PBST three times for 10 min each and secondary antibodies were applied for 2 h at room temperature. The following primary antibodies were used in these studies: p63 mouse IgG (clone 4A4, Santa Cruz Biotechnology, sc8431, 1:100) or rabbit IgG (GenTex, GTX102425, 1:300), CK5 rabbit IgG (Covance, AF-138, PRB-160P, 1:300), or chicken IgY (Covance, SIG-3475, 1:300), Ck20 mouse IgG2a, kappa, clone Ks20.8 (Dako, M7019, 1:250), Ki67 rabbit IgG (Abcam, ab15580, 1:300), Tom20 rabbit IgG (Santa Cruz, SC-11415, 1:2000), E-cadherin goat IgG (R&D System, AF748, 1:300), PPARG rabbit IgG (Cell Signaling Technology, #2435, 1:100), FABP4 goat IgG (R&D Systems, AF1443, 1:1000), Krt14 chicken IgY (BioLegend, 906001, 1:500) or rabbit IgG (BioLegend, 905301, 1:300), ZO1 rabbit IgG (Thermo Fisher Scientific, 40-2300, 1:100), Krt6a rabbit (LSbio, LS-B12036, 1:500), Krt10 mouse IgG1 (Santa Cruz, Sc-53252, 1:500), CLDN8 rabbit IgG (GeneTex, GTX77832, 1:50), UCHL1 mouse IgG1 (Santa Cruz, sc-271639, 1:200), SNAIL+SLUG rabbit IgG (Abcam, ab180714, 1:200), laminin rabbit (Sigma, L9393, 1:100), actin, a smooth muscle mouse (Sigma, C6198, 1:300), NF-κB p65 rabbit IgG (Abcam, ab19870, 1:300), CD45 rat IgG2b (BD Sciences, 550539, 1:50), F4/80 rat IgG2a (Thermo Fisher Scientific, 14-4801-82, 1:100), CD3 rat IgG1 (Abcam, ab11089, 1:50), CD19 rat IgG2a (Thermo Fisher Scientific, 14-0194-82, 1:500), SPRR1A rabbit IgG (Biorbit, orb1053, 1:200), COX1 rabbit (Novus, NBP1-85500, 1:200), SOD1 mouse IgG (Santa Cruz, sc-17767, 1:200), SOD2 rabbit IgG (Abcam, ab13533, 1:500), and CPT2 rabbit IgG (Genetex, GTX33117, 1:50). UPK1A, UPK1B, and UPK2 antibodies are gifts kindly provided by Dr. Tung-Tien Sun at NYU. The following secondary antibodies from Jackson Immunoresearch were used in our studies: Alexa Fluor 488 donkey anti-rabbit IgG (711-545-152; 1:600), Alexa Fluor 488 donkey anti-mouse (715-545-150;1:600), CY3-conjugated donkey anti-rabbit IgG (711-165-152; 1:600), Alexa Fluor 594 donkey anti-mouse IgG (715-585-151; 1:500), Alexa Fluor 647-conjugated donkey anti-mouse IgG (715-605-150; 1:300), Alexa Fluor 647-conjugated donkey anti-rabbit IgG (711-605-152; 1:300), Alexa Fluor 488 donkey anti-chicken (703-545-155, 1:500), Alexa Fluor 594 donkey anti-chicken (703-585-155, 1:500), BODIPY 493/593 (Thermo Fisher Scientific D-3921, 4 μg/ml), MitoTracker™ Red CMXRos (Thermo Scientific M7512, working solution 25 nM), WGA, and Alexa Fluor™ 488 conjugate (Thermo Fisher Scientific W11261, 10 μg/ml). DAPI (4′,6-diamidino-2-phenylindole) was either applied as part of the secondary antibodies cocktail or for 10 min, for nuclear staining, and then the slides were sealed with coverslips.

**Representative images for figures.** For adult samples, at least two slides (eight sections) were analyzed from each marker or set of markers from at least three bladder samples. Five images were generated per/section and representative images were chosen. For adults, sections were separated by 50 μm. For embryonic samples, each set instance of marker analysis was performed on six sections from three or more bladder samples, and sections were separated by 35 μm. At least five images were taken from each section and a representative image was chosen for the figure.

**Statistical analysis.** All quantitation was performed on at least three independent biological samples, using the ImageJ software. Data presented are mean values ± s.e.m. Statistical analysis was performed using the GraphPad Prism software v8. In two group comparisons, statistical significance was determined using a two-tailed Student's $t$ test, considering a value of $p < 0.05$ as significant. Multiple comparisons were performed using the Kruskal–Wallis statistical test. All sample sizes met the minimum requirements of the respective statistical test used. The number of samples used in the experiments is included in figure legends.

**Fluorescent microscopy.** Immunofluorescence images were collected using a Zeiss Axiovert 200M microscope with an Apotome (Zeiss). Bright-field images were collected using a Nikon Eclipse TE200 microscope. Confocal microscopy was performed on an A1R MP confocal microscope (Nikon Instruments) and data were analyzed and rendered using NIS Elements (Nikon) and the Fiji package of ImageJ.

**Electron microscopy.** Fixed samples were osmicated 1–2 h with 1.5% (w/v) reduced $OsO_4$ in 100 mm cacodylate, pH 7.4, washed several times with distilled water, and then block stained overnight at 4 °C in 0.5% (w/v) aqueous uranyl acetate (Electron Microscopy Sciences, Hatfield, PA). Tissues were dehydrated in a graded series of ethanol, embedded in the epoxy resin LX-112 (Electron Microscopy Sciences), and sections (pale gold in color) were cut with a Diatome diamond knife (Electron Microscopy Sciences). Sections were counterstained with uranyl acetate and lead citrate and viewed on a JEOL 1011 transmission EM with a side mount AMT 2K digital camera (Advanced Microscopy Techniques, Danvers, MA). Images were imported into Photoshop CC (Adobe, San Jose, CA), adjusted for brightness and contrast, and then assembled in Adobe Illustrator CC. All of the EM studies were performed using an $n = 3$ for experimental and control. While our study does not permit the quantification of the frequency of each phenotype (mitobodies, and junctional defects), these abnormalities were readily observed in every mutant analyzed. None of these features were observed in any of the controls. The degree to which each phenotype represents the displayed image in the figures is described below for the specific image.

## Data availability

Data that support the findings of this study have been deposited in Gene Expression Omnibus database under the accession code GSE123779.

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

## Acknowledgements

We thank Daniel Metzger for the Krt5[CreER]T2 line, Ira Goldberg for the Pparg[fl/fl] mice, Molly Ingersoll, Sankar Ghosh, William Kim, Henry Sun, Xue-Rue and David Degraff for discussions and critical reading of the manuscript, the Herbert Irving Comprehensive Cancer Center Flow Cytometry Core, the Confocal and Specialized Microscopy Core, and the Molecular Pathology Core. This work was supported by: DK095044 and U01 DK0945300 (C.L.M./S.P.), TJMCU508-5926-URO (C.L.M.), and a T32 Training Grant DK07328 (to T.T.). EM was performed at the University of Pittsburgh with a JEOL 1011 transmission EM 1S10RR 01900 (PI, Simon Watkins).

## Author contributions

C.L. designed experiments and analyzed the phenotypes in *Pparg* mutant mice and prepared figures for the paper, T.T. assisted with analysis of embryonic and adult phenotypes and helped write the paper, E.B. and H.K. performed immunostaining to validate results of RNA-Seq experiments, S.T.T. performed EM on Pparg mutant tissue; T.X. assisted with writing of the paper; M.P. assisted in analyzing mitochondria inclusions in *Pparg* mutant; M.R., K.S., M.T. assisted with HE/IHC analysis of mutant phenotypes, S.P., M.A., C.L., X.C. and J.H. assisted with analysis of RNA-Seq data, C.L.M. helped interpret results, design experiments, and wrote the paper.

## Additional information

**Competing interests:** The authors declare no competing interests.

