## [Peer Review File · Nature Communications]

Reviewers' Comments:

Reviewer #1:

Remarks to the Author:

The authors present a rather elegant series of experiments in conditional knockout murine models of urothelial cell differentiation coupled with extensive fluorescent immunohistochemistry staining of specific mechanistic genes of interest and RNAseq transcriptome analyses. Overall, the authors present convincing evidence of the critical role that PPARG plays in differentiation of the normal superficial S-cells, that loss/deletion of PPARG shifts the urothelium toward a basal cell dominated composition with increased squamous features and deranged mitochondrial structure and gene expression. The authors are requested to address the following questions/concerns.

1) Since all of the data presented in this manuscript compares protein and gene expression differences between wild type and conditional knockout mutant mice, an obvious question that has important implications for the impact of this work is whether the changes presented persist over time. In the methods section, it appears that the majority of the mice (with the exception of the neonatal mice) were 8-12 weeks old at the time tamoxifen was administered to knockout PPARG, that they received tamoxifen for 5-7 days, and then were sacrificed to examine the effects on urothelial cell differentiation. The authors specifically conclude in their discussion that loss of PPARG function alone is not sufficient to promote urothelial/squamous cell carcinoma. But would you expect to see actual carcinoma tumors form only 5-7 days after PPARG knockout? Have the authors followed these PPARG knockout mice out to long-term time points and seen no further phenotypic or gene expression changes? If so, this data should be provided in the supplementary information and a comment should be added somewhere in the manuscript to reference this. If not, then the conclusion about the lack of carcinogenic potential from loss of PPARG function should be tempered and the absence of long-term follow up of these mice should be acknowledged as a limitation.

2) Similarly, in the UPEC infection tissue injury work, it is stated that the RNASeq analyses were performed 24 hrs, 72 hrs, and 4 weeks post-infection. These seem like completely reasonable time points, but a justification should be provided for how they were chosen. Specifically, the long gap between the 72 hour time point and the 4 week time point allows for the possibility to miss important mechanistic mediators of the inflammatory injury response that may have resolved back to baseline by the 4 week time point. It's impossible to sample every possible time point, but one could argue that an intervening time point between the 72 hour and the 4 week time point (e.g. 2 week time point) could have provided valuable insights. The authors are asked to justify the choice of time points analyzed / not analyzed.

3) In the statistical methods section, the authors state that comparisons were made with n=3 or greater animals per genotype. Three mice seems like a very low number. How many comparisons were made with only three mice per genotype? Was it only an outlier analysis that was not feasible to replicate with additional mice, or were most analyses done with only 3 mice per genotype? If only 3 mice per genotype was the majority of comparisons, then please provide some justification for the small sample size and explanation for how such small sample sizes can provide statistical confidence in the data generated from them.

Noah M. Hahn, MD

Johns Hopkins University Greenberg Bladder Cancer Institute
Baltimore, Maryland

Reviewer #2:

Remarks to the Author:

This manuscript provides compelling evidence regarding an important role for PPARG in the

maintenance of urothelial differentiation in adult mice, as well as for urothelial development and regeneration. Perhaps most interestingly (and significantly), the work links deficits in Pparg expression and function to deficits in mitochondrial activity. This work is significant and will be of interest to individuals both within the field of urothelial cell biology, as well as those outside the field, including members of the steroid hormone receptor and developmental biology fields.

Minor concerns

1. It is stated that S cells express the highest level of Pparg. Was this quantified? It appears that other cell populations (I cells?) express Pparg.
2. In 3rd paragraph of the results section the authors state "urothelial cells lining the luminal layer were small compared to controls (Figure 1 I,m,n)." Which cell populations specifically are the authors referring to? These images are quite small and difficult to interpret fully.
3. Figure 1R is not discussed explicitly in the text.
4. In the text it would be helpful if the authors explicitly state if the model is an inducible system (i.e., in areas it is not explicitly stated that the Krt5-Cre ERT2 system is tamoxifen inducible).
5. Figure 2B is discussed first in the text relative to a. The figure should be re-arranged to accommodate the flow of the text, which is quite nice.
6. In the introduction, I thought urothelial cells extend to the proximal urethra, and not just the bladder neck. Is this not the case?
7. PPARG is amplified in a subset of bladder cancers as well, and there are recent studies by Bernard-Pierrot's group (PMID 30604486) that might be worth citing.
8. Were markers of apoptosis additionally examined in the Shh-cre experiments?
9. It would be interesting to see a heatmap showing altered transcription factors in Figure 1.
10. In the section beginning "Analysis of ShhCre;Ppargfl/fl mutants in which Pparg is deleted... were rapidly inactivated in mutants (Fig 3a,b)." I think the authors are referring to 3E, F. Please double check. Also at the end of this page, instead of 3I should it be 3L?
11. What is "over-representation analysis"?
12. What is the definition of an "immature S cell"? Not completely differentiated?
13. Was there any sex differences in the phenotypes in any of the experiments?
14. Are there differences in mitochondrial metabolism in subsets of human bladder cancer included in the TCGA or other cohorts?

Major concerns

1. It is suggestive that luminal markers such as Krt20 are detected at reduced levels in the Shh-Cre mice experiments (Figure 1). However, it is not clear if these findings are reflective of decreased Krt20 expression, or the fact that pparc KO is toxic to urothelial cells, thus resulting in cell death (and inability to detect Krt20). Is Krt20 completely undetectable? Some of this might be addressed by subsequent experiments shown in the current submission, but I am not sure.
2. The orthogonal approaches shown in Figure 2 (mRNA expression and IF) are a clear strength, but this data should be quantified. While trends are important and support the overall conclusions, I am curious to know how many of these differences are statistically significant. Also-can the IF be quantified (counting positive/negative cells)?
3. While it is almost certainly safe to conclude (based on the findings) that Pparg regulates mitochondrial function in a cell autonomous manner, I am not sure that the same can be said about the cell autonomous manner in which pparc regulates cell differentiation or survival. This is true because it is possible that Pparg ablation influences barrier integrity, resulting in urine leaking through the urothelium, and potentially adversely affecting cell differentiation or survival. Therefore this conclusion should be scaled back a bit.
4. Figure 4 (Krt5-CreERT2/Pparg KO). It is difficult to judge the extent of squamous differentiation. Indeed, I would argue that the upregulation of a few markers of squamous differentiation in isolated cell populations is not indicative of over squamous differentiation (although it might be indicative of early stages of this process). Is hyperkeratinization and/or presence of intercellular bridges observed?

Reviewer #3:

Remarks to the Author:

Liu and co-workers used different cell type-specific inducible PPARgamma knockout mouse models to study the role of PPARgamma in urothelium, both under healthy and UTI conditions. Their data suggest that PPARgamma is needed for normal urothelium development and homeostasis, for mitochondrial function, and is a potent regulator of the inflammatory response after UTI.

Major concern:

The authors study the effect of PPARgamma deficiency by using the Cre-Lox model. They use 3 different tamoxifen-inducible Cre mouse lines to drive PPARgamma depletion in distinct urothelial sub-populations. In each case, they use the PPARg fl/fl as control mice. The authors need to prove that these mice are suitable control mice for these studies. Numerous studies have shown that Cre-recombinase expression by itself can have effects on cell differentiation and other aspects evaluated in this manuscript. Therefore, the PPARg fl/fl mice might be the wrong control mice. Have the authors, or others, studied the 3 cre-expressing lines and do they have proof that none of the observed phenotypes are (partially) due to Cre-expression? Moreover, while in the KRT5CreERT2 and UPK2CreERT2 mice the Cre recombinase is expressed as a transgene, in the ShhCre mice the Cre gene actually replaces the endogenous Shh gene (knock in). In the latter case, since Shh has been shown to play a critical role in bladder development (see PMID 26757905), the authors need to provide proof that having only one functional Shh allele does not affect the parameters evaluated in this study. In short, the authors either have to show data from the 3 Cre-expressing lines as controls throughout their study, or be able to cite previous studies (from others) that show that the 3 Cre-expressing mouse strains behave identical to wild-type mice in regards to the parameters studied.

Other concerns:

1) The role of PPARgamma in bladder cancer is very controversial, with numerous studies both claiming an oncogenic or anti-cancerous role for PPARgamma. This is not evident from the way the manuscript is currently written. The short paragraphs in the introduction, results, and discussion that touch upon this subject are vague (using terms like "associated with" and "regulate") and cite few recent publications. The authors should develop this further and more clearly (in a neutral manner), and include recent studies (examples PMIDs 30651555, 30912275, 30845932).

2) In the results (page 10), it is stated that "Upk3, which labels I-cells and S-cells, and Krt20 which labels mature S-cells were both down-regulated (Fig. 3a-i)". However, Fig. 3i shows an increase in S cells. Please explain.

3) PPARgamma can regulate NFkappaB through different mechanisms, including by sequestering the P65 subunit, but also by regulating I-kappa-B-alpha and IKK-alpha/beta levels. It is not clear what is stained as NFkappaB in Fig. 5 and Supplemental Fig. 4 (no description in Methods section). The whole part on NFkappaB should be developed in more detail, including mechanistic details.

4) Supplement Fig. 3 is never mentioned in the text.

Minor concerns:

The manuscript has been written in a very sloppy manner and includes numerous errors of which a non-exhaustive list follows below. Especially the Figure legends contained many errors and the Methods section is far from complete. Even though these are minor concerns, the authors could have made a bigger effort to submit a version that doesn't come across as an early draft.

1) Introduction (page 5): "Pparg expression is down-regulated in the basal subtype of BC, which has squamous features". It is not clear what BC refers to; bladder cancer or basal cells.

2) Results (page 7): "Pparg expression is undetectable in the Basal cell compartment until E18

(Fig. 1g, green arrow).” Fig. 1g shows situation at E16, not E18.

3) Page 9: “However, the mitochondrial membrane is impermeable to fatty acids and a specialized carnitine carrier system consisting of Cpt1, Slc25a20 and Cpt2 controls fatty acid transport (Spinelli and Haigis, 2018). We observed down-regulation of all 3 genes in ShhCre; Ppargfl/fl mutants (Fig. 2b,e). In addition, 15 genes that encode members of complex 1 NADH ubiquinone oxidoreductase were down-regulated, including Nd4, Nd5 and Nd6 that are transcribed from mitochondrial DNA (Fig. 2b; Cpt2 immunostaining is shown in Fig. 2e)”

It would make more sense to write: “However, the mitochondrial membrane is impermeable to fatty acids and a specialized carnitine carrier system consisting of Cpt1, Slc25a20 and Cpt2 controls fatty acid transport (Spinelli and Haigis, 2018). We observed down-regulation of all 3 genes in ShhCre; Ppargfl/fl mutants (Fig. 2b, Cpt2 immunostaining is shown in Fig. 2e). In addition, 15 genes that encode members of complex 1 NADH ubiquinone oxidoreductase were down-regulated, including Nd4, Nd5 and Nd6 that are transcribed from mitochondrial DNA (Fig. 2b)”

4) Page 9: “Fig. S2 b,c” Be consistent and write Supplementary Fig. 2 b,c

5) Page 10: “cell population in ShhCre; Ppargfl/fl mutants (Fig. 1h,i,k,o; Supplementary Fig. 1k-n).” Should be “Fig. 1h,l,k,o”.

6) Page 12: “an possibility” should be “a possibility”.

7) Page 12: “may normally act in basal cells suppress squamous” should be “basal cells to suppress”.

8) Page 13: All referrals to Supplementary Fig. 5 in first paragraph should actually refer to Supplementary Fig. 4.

9) Page 13: “phenotype that continuing after infection was cleared (Fig. 5a-i)” should be “phenotype that continued after infection was cleared (Fig. 5f-h,i)”.

10) The Methods section does not describe the mTmG mice used for Supplementary Fig. 5. Furthermore, the section on immunostaining only lists a fraction of the primary antibodies used for the stainings shown. Missing are PPARgamma, FABP4, Krt14, CD19, CD3, F4/80, CD45, NFkappaB, SOD1/2, Cpt2, Cox1, etc....

11) Figure 2 legend: “(f-h) Immunofluorescence staining of superficial cell markers (f) Cldn8, (g) Sprr1a and (h) Uchl1” should be “(i-k) Immunofluorescence staining of superficial cell markers (i) Cldn8, (j) Sprr1a and (k) Uchl1”

12) Supplement Fig. 2 legend: arrows indicating vesicles are not mentioned.

13) Figure 3 legend: “Co-staining of (a, e) Krt20 and P63, (b, f) Fabp4 and Upk3a, (c, g) Ki67 and Upk3a, (d, h)” should be “Co-staining of (a, e) Krt20 and PPARgamma, (b, f) Fabp4, P63, and Upk3a, (c, g) Ki67, P63, and Upk3a, (d, h)”.

14) Figure 3 legend: “numbered from the smallest inclusion (1) to the largest inclusion (4).” No such numbering is visible in the figure.

15) Figure 4 legend: “(a-d) Transmission electron microscopy image of the urothelium from adult Ppargfl/fl controls or Krt5CreER; Ppar fl/fl mutants.” Should be “ShhCreER; Ppar fl/fl”

16) Figure 5 legend: Panel i is not mentioned/described.

Rebuttal

Thank you, reviewers, for your careful reading and comments. Below, we address each point raised by each reviewer:

Reviewer 1:

1. The authors specifically conclude in their discussion that loss of PPARG function alone is not sufficient to promote urothelial/squamous cell carcinoma. But would you expect to see actual carcinoma tumors form only 5-7 days after PPARG knockout? Have the authors followed these PPARG knockout mice out to long-term time points and seen no further phenotypic or gene expression changes? If so, this data should be provided in the supplementary information and a comment should be added somewhere in the manuscript to reference this.

Response: We have included data showing urothelial differentiation in *ShhCre;Pparg^{fl/fl}* mutants at 2 weeks, 5 months and 1 year after Tamoxifen induction (now included Supplementary Figure 5). We observe abnormal differentiation in the basal compartment, including an expanded K14-Basal population, however we have not observed signs of invasion or tumors in these animals.

2) Similarly, in the UPEC infection tissue injury work, it is stated that the RNA-Seq analyses were performed 24 hrs, 72 hrs, and 4 weeks post-infection. These seem like completely reasonable time points, but a justification should be provided for how they were chosen.

Response: We have analyzed urothelial regeneration in *ShhCre;Pparg^{fl/fl}* mutants and controls at 12h, 24h, 72h, 2 weeks, 4 weeks, 6 weeks and 1 year post-infection using the UTI model. We chose the 24h and 72h time-points for comparison because 24h represents the peak of urothelial proliferation post-infection and 72h is the time-point when newly formed Superficial cells are observed in the luminal layer in controls. Superficial cells when formed do not immediately express Krt20; they undergo maturation, which takes about 2 weeks, at which time Krt20 is detectable throughout the luminal layer (which is why we use the 2-week time point). There is a transient increase in the K14-Basal cell population in both controls and mutants during regeneration. Urothelial differentiation returns to normal in controls but in mutants, K14-basal cell expansion continues, and is accompanied by expression of squamous markers, a condition that persists for a year or more. We chose the 1 month time point versus 2 weeks as a representative stage for comparison to maximize the possible differences in gene expression between controls and mutants. In summary, we have not observed signs of tumor formation in these animals after a year or more, either during homeostasis or after UTI. This data is included in Supplementary Figure 5.

3) In the statistical methods section, the authors state that comparisons were made with n=3 or greater animals per genotype. Three mice seems like a very low number. How many comparisons were made with only three mice per genotype? Was it only an outlier analysis that was not feasible to replicate with additional mice, or were most analyses done with only 3 mice per genotype? If only 3 mice per genotype was the majority of comparisons, then please provide some justification for the small sample size and explanation for how such small sample sizes can provide statistical confidence in the data generated from them.

Response: This is a good point. We have actually analyzed a *minimum* of 3 samples for each experiment, however we generally use more than 3 animals in a cohort. The numbers of animals analyzed are now included in figure legends for each experiment. In most cases, we determine phenotypes by counting cells expressing a given marker on many sections from mutants and controls which gives the analysis considerable power. With the exception of RNA-seq data, where we analyzed 5 or more samples, we did not observe significant variability in the phenotypes examined in this manuscript.

Reviewer 2

It is stated that S cells express the highest level of *Pparg*. Was this quantified? It appears that other cell populations (I cells?) express *Pparg*.

Response: During homeostasis, *Pparg* expression is detected throughout the urothelium, however expression is consistently much higher in S-cells compared to I-cells and Basal cells (Figure 1, Panels A and C). Consistent with this,

Fabp4, a direct target of *Pparg* is present at high levels in S-cells and is expressed at much lower levels in the I-cell and Basal populations (Panel B).

In 3rd paragraph of the results section the authors state “urothelial cells lining the luminal layer were small compared to controls (Figure 1 l,m,n).” Which cell populations specifically are the authors referring to? These images are quite small and difficult to interpret fully.

Figure 1. *Pparg* and *Fabp4* expression. *Pparg* expression and *Fabp4* expression are highest in S-cells.

Pparg and Fabp4 expression in the adult urothelium. **A.** *Pparg* (green) and *Krt20* (red) stained adult wild type urothelium. *Pparg* is expressed in basal cells (green arrow) intermediate cells (purple arrow) and Superficial cells (yellow arrows). **B.** Immunostaining showing expression of *P63* (purple) and *Fabp4*, a transcriptional target of *Pparg* which is expressed at high levels in superficial cells (yellow arrows). **C.** *Pparg* staining in basal cells (green arrow), Intermediate cells (purple arrow) and Superficial cells (yellow arrows). Note that expression is present in all cell types, with highest levels in the Superficial layer. The Inset in (C) shows negative control for *Pparg* staining (no primary antibody).

Response: We have increased the resolution of the Figure 1h-o (see Below, Figure 2) so that it is easier to see the relative sizes of cells and expression of markers. The mutant and control Superficial cells in Fig1 i,j,m,n are highlighted with dashed white lines which is now indicated in the text. Work from Henry Sun's lab indicates that loss of *Upk2* or *Upk3* results in a reduction in the size of superficial cells. [(Kong XT1, Deng FM, Hu P, Liang FX, Zhou G, Auerbach AB, Genieser N, Nelson PK, Robbins ES, Shapiro E, Kachar B, Sun TT. (2004) Roles of uroplakins in plaque formation,

Figure 2. New, higher magnification Fig. 1h-o showing ZO1, Krt20, basal markers and sizes of S-cells in mutants and controls (dashed white lines).

umbrella cell enlargement, and urinary tract diseases. *J Cell Biol.* 20, p1195-1204.]. The luminal layer of superficial cells is lined by crystals that are assembled from Upks which are transported from the Golgi in specialized vesicles to the apical membrane, where they fuse, increasing the surface area of the cell. *Pparg* mutants produce very low quantities of Upks which are essentially biomass, a possible explanation for the relatively small size of the Superficial cells in mutants.

3. Figure 1R is not discussed explicitly in the text.

Response: We have now included 1r in the text describing the data in Figure 1.

4. In the text it would be helpful if the authors explicitly state if the model is an inducible system (i.e., in areas it is not explicitly stated that the *Krt5-Cre ERT2* system is tamoxifen inducible).

Response: For the *Krt5* line, we have added to the text: “Analysis of adult *Krt5CreERT2;Pparg^{fl/fl}* mutants 14 days after Tamoxifen induction revealed down-regulation of *Pparg* in basal cells, while the expression level remained the same in S-

cells and I-cells (Fig. 4e,h). Immunostaining showed basal cell abnormalities in *Krt5CreERT2;Pparg^{fl/fl}* mutants similar to those observed in *ShhCre;Pparg^{fl/fl}* mice, including an expanded K14-basal population and up-regulation of Krt10 (Fig. 4e-j).” For the *UP2CreERT2* line, we have added to the text: “To begin to address this, we used the Tamoxifen inducible *Upk2CreERT2* line³⁴ to selectively delete *Pparg* in S-cells and I-cells, then we analyzed the effects on urothelial homeostasis.”

5. Figure 2B is discussed first in the text relative to a. The figure should be re-arranged to accommodate the flow of the text, which is quite nice.

Response: We have changed the order of 2a and 2b components in Figure 2 as suggested.

6. In the introduction, I thought urothelial cells extend to the proximal urethra, and not just the bladder neck. Is this not the case?

Response: The proximal bladder neck/trigone is a transition zone between the bladder and urethra. There is some overlap between urothelial and urethral markers at this site, however it is not clear how far the physical features of the urothelial barrier (high-resistance tight junctions and apical plaque) extend into the proximal urethra/prostatic urethra in males.

7. PPARG is amplified in a subset of bladder cancers as well, and there are recent studies by Bernard-Pierrot’s group (PMID 30604486) that might be worth citing.

Response: We have replaced the text with a new paragraph describing the potential role for *Pparg* in bladder cancer subtypes, citing the work from Bernard-Pierrot’s group: “Mapping of the mutational landscape of muscle-invasive bladder cancers (MIBC) together with unsupervised clustering analysis of the whole genome expression data revealed that MIBC can be sub-categorized into luminal and basal subtypes. These subtypes are histologically distinct and display discrete sets of mutations and gene expression signatures¹⁴⁻¹⁹. These analyses reveal alterations in *PPARG* expression and signaling, suggesting that *PPARG*-dependent transcriptional regulation may be important in the etiology of urothelial carcinoma. Supporting this, *PPARG* copy number expansion and increased expression of *FABP4*, a direct *PPARG* transcriptional target, were observed in luminal-tumors²⁰⁻²². Activating mutations in *PPARG* and *RXRA*, a *PPARG* binding partner, were also observed in luminal MIBC^{23,24}. In addition, upregulation of these gene sets important for lipid metabolism and adipogenesis in patients that harbor *PPARG* gain-of function mutations suggest that *PPARG* may be an important regulator of lipid metabolism in the luminal subtype of MIBC.

The exact contribution of *PPARG* to the etiology of the basal subtype of urothelial carcinoma is less clear. *PPARG* expression is low in basal subtype tumors compared to healthy urothelium, and *PPARG* is down-regulated in Claudin-low tumors, which have basal-like features. Interestingly, genes encoding cytokines and chemokines are up-regulated in Claudin-low basal-like tumors, which may reflect unregulated NF-κB signaling due to low levels of *PPARG*²⁵. Expression of *PPARG* and its binding partner *RXRA*, are reduced in the SCC-like (SCCL) subtype of MIBC, which shares many features with the basal subtype, including gene expression signatures and common mutations. Transcriptional analysis of those basal or basal-like subtype of tumors revealed a large cluster of genes important for lipid metabolism that were down-regulated compared to the luminal subtype of tumors. *In silico* Chip-Seq analysis revealed that many of these down-regulated genes contained *PPARG* binding sites in their regulatory regions, suggesting that these are *PPARG*-transcriptional targets²⁶.”

8. Were markers of apoptosis additionally examined in the *Shh-cre* experiments?

Response: We performed TUNEL staining and immunostaining for activated caspase 3 at several stages after Tamoxifen induction but we did not detect evidence for apoptosis.

9. It would be interesting to see a heat map showing altered transcription factors in Figure 1.

Response: The heat map showing changes in expression of Superficial cell markers and transcription factors in *ShhCre;Pparg^{fl/fl}* mutants is included in Figure 2, which also includes basal cell markers.

10. In the section beginning “Analysis of *ShhCre;Pparg^{fl/fl}* mutants in which *Pparg* is deleted... were rapidly inactivated in mutants (Fig 3a,b).” I think the authors are referring to 3E, F. Please double check. Also, at the end of this page, instead

of 3I should it be 3L?

Response: Here is the corrected paragraph: "indicating that both *Pparg* expression and signaling were decreased in mutants compared to controls (Fig. 3a,b,e,f). This analysis also revealed that *Upk3a*, which is highly enriched in S-cells, and *Krt20*, which labels mature S-cells, were both down-regulated (Fig. 3a-h; Supplementary Fig. 3a-f), suggesting that *Pparg*-dependent transcription is important for proper maintenance of S-cells." In addition, we replaced incorrect labeling of 3i with 3l.

11. What is "over-representation analysis"?

Response: Over-representation analysis (OAR) was performed using gene set analysis with the ConsensusPathDB-mouse database (<http://cpdb.molgen.mpg.de/MCPDB>). We uploaded lists of genes from RNA-seq analysis of *ShhCre;Pparg^{fl/fl}* mutant urothelium with p-values of 0.05 or higher, which were 1.5-fold up or down-regulated. For OAR, we used a p-value of 0.01 for analysis. According to the website: "The gene identifiers are mapped to physical entities in ConsensusPathDB. Over-represented sets are searched among currently three categories of predefined gene sets: network neighborhood-based sets, pathway-based sets and Gene Ontology-based sets."

12. What is the definition of an "immature S cell"? Not completely differentiated?

Response: Despite the lack of expression of *Krt20*, a marker of mature S-cells, and low levels of *Upk* family members, which are highly expressed in wild type S-cells, we use the term "Immature S-cells" since: (i) they express *ZO1*, which is specifically expressed in tight junctions of S-cells, not in I-cells; and because S-cells in mutants lack detectable expression of *p63*, which marks I-cells.

13. Was there any sex differences in the phenotypes in any of the experiments?

Response: We did not observe sex differences in our experiments.

14. Are there differences in mitochondrial metabolism in subsets of human bladder cancer included in the TCGA or other cohorts?

Response: Eriksson et al (Eriksson, P., Aine, M., Veerla, S., Liedberg, F., Sjobahl, G., and Hoglund, M. (2015). Molecular subtypes of urothelial carcinoma are defined by specific gene regulatory systems. *BMC Med Genomics* 8, 25) identified a cluster of genes down-regulated in the LUND cohort, that regulate lipid metabolism in the Basal and SCCL subtypes of urothelial carcinoma. These were not down-regulated in luminal or UroA tumors. This has been included in the introduction: "Expression of *PPARG* and its binding partner *RXRA*, are reduced in the SCC-like (SCCL) subtype of MIBC, which shares many features with the basal subtype, including gene expression signatures and common mutations. Transcriptional analysis of those basal or basal-like subtype of tumors revealed a large cluster of genes important for lipid metabolism that were down-regulated compared to the luminal subtype of tumors. *In silico* Chip-Seq analysis revealed that many of these down-regulated genes contained *PPARG* binding sites in their regulatory regions, suggesting that these are *PPARG*-transcriptional targets²⁶."

Major concerns

1. It is suggestive that luminal markers such as *Krt20* are detected at reduced levels in the *Shh-Cre* mice experiments (Figure 1). However, it is not clear if these findings are reflective of decreased *Krt20* expression, or the fact that *pparg* KO is toxic to urothelial cells, thus resulting in cell death (and inability to detect *Krt20*). Is *Krt20* completely undetectable? Some of this might be addressed by subsequent experiments shown in the current submission, but I am not sure.

Response: *Krt20* is not detectable in S-cells of *ShhCre;Pparg^{fl/fl}* mice, we hypothesize because they fail to mature. Supporting this, we do observe expression of other markers including *ZO1*, which is specifically expressed in S-cells, as well as *Gata3* and *Foxa1* (please see below).

2. The orthogonal approaches shown in Figure 2 (mRNA expression and IF) are a clear strength, but this data should be quantified. While trends are important and support the overall conclusions, I am curious to know how many of these differences are statistically significant. Also-can the IF be quantified (counting positive/negative cells)?

Response: p-values for each of the genes shown as up or down-regulated in Figure 2 are now included in Table 1. All have a p-value ≤ 0.001 , with the exception of *Atp5s*, *Mcee*, *Ndufa12*, *Cpt1a* and *Gata3* which have a p-value ≤ 0.05 . P-values for changes in expression of *Krt5*, *Krt6b*, *Foxa1* and *Ucp*, p-values were not statistically significant (0.08, 0.09, 0.18 and 0.4, respectively, however down-regulation of *Krt5* and *Foxa1*, and up-regulation of *Krt6b* were validated with immunostaining.

3. While it is almost certainly safe to conclude (based on the findings) that *Pparg* regulates mitochondrial function in a cell autonomous manner, I am not sure that the same can be said about the cell autonomous manner in which *pparg* regulates cell differentiation or survival. This is true because it is possible that *Pparg* ablation influences barrier integrity, resulting in urine leaking through the urothelium, and potentially adversely affecting cell differentiation or survival. Therefore, this conclusion should be scaled back a bit.

Response: Figure 3 shows expression *Gata3*, *Foxa1* and *ZO1* in S-cells from mutants versus controls. The robust expression of *Gata3* and *Foxa1*, and specific expression of *ZO1* in S-cells is evident in both mutants and controls, suggesting that S-cells are still viable in the mutants, despite the loss of *Krt20* and other urothelial markers. We have not observed differences in permeability between mutants and controls using methylene blue staining, but we have not performed more exhaustive assays of permeability; hence we will tone down the text as follows: **The title of the section "*Pparg* regulates differentiation, mitochondrial functions and survival of S-cells in a cell autonomous manner" is now: "*Pparg* regulates mitochondrial functions in S-cells."**

4. Figure 4 (*Krt5-CreERT2/Pparg* KO). It is difficult to judge the extent of squamous differentiation. Indeed, I would argue that the upregulation of a few markers of squamous differentiation in isolated cell populations is not indicative of over squamous differentiation (although it might be indicative of early stages of this process). Is hyperkeratinization and/or presence of intercellular bridges observed?

Figure 3. Gene expression in adult luminal S-cells in *ShhCre;Pparg^{fl/fl}* mice and controls. Left Panel: *Gata3* staining in luminal S-cells in controls and mutants. Middle Panel: *Foxa1* expression in luminal cells in controls and *ShhCre;Pparg* mutants. Right panel: *ZO1* expression in luminal S-cells of controls and *ShhCre;Pparg^{fl/fl}* mice.

Response: Point well taken. We have changed the text from : "*Analysis of adult Krt5CreERT2;Pparg^{fl/fl} mutants 14 days after Tamoxifen induction revealed that Pparg was selectively down-regulated in Basal cells, and mutants displayed*

squamous differentiation in a similar pattern as observed in *ShhCre;Pparg^{fl/fl}* mice including an expanded K14-basal population and up-regulation of squamous markers such as Krt10, that are not present in the healthy urothelium" to "Analysis of adult *Krt5CreERT2;Pparg^{fl/fl}* mutants 14 days after Tamoxifen induction revealed down-regulation of *Pparg* in basal cells, while the expression level remained the same in S-cells and I-cells (Fig. 4e,h). Immunostaining showed basal cell abnormalities in *Krt5CreERT2;Pparg^{fl/fl}* mutants similar to those observed in *ShhCre;Pparg^{fl/fl}* mice, including an expanded K14-basal population and up-regulation of Krt10 (Fig. 4e-j)."

Reviewer #3

Liu and co-workers used different cell type-specific inducible PPARgamma knockout mouse models to study the role of PPAR gamma in urothelium, both under healthy and UTI conditions. Their data suggest that PPARgamma is needed for normal urothelium development and homeostasis, for mitochondrial function, and is a potent regulator of the inflammatory response after UTI.

Major concern:

The authors study the effect of PPARgamma deficiency by using the Cre-Lox model. They use 3 different tamoxifen-inducible Cre mouse lines to drive PPARgamma depletion in distinct urothelial sub-populations. In each case, they use the PPARg fl/fl as control mice. The authors need to prove that these mice are suitable control mice for these studies. Numerous studies have shown that Cre-recombinase expression by itself can have effects on cell differentiation and other aspects evaluated in this manuscript. Therefore, the PPARg fl/fl mice might be the wrong control mice. Have the authors, or others, studied the 3 cre-expressing lines and do they have proof that none of the observed phenotypes are (partially) due to Cre-expression? transgene, in the *ShhCre* mice the Cre gene actually replaces the endogenous *Shh* gene (knock in). In the latter case, since *Shh* has been shown to play a critical role in bladder development (see PMID 26757905), the authors need to provide proof that having only one functional *Shh* allele does not affect the parameters evaluated in this study. In short, the authors either have to show data from the 3 Cre-expressing lines as controls throughout their study, or be able to cite previous studies (from others) that show that the 3 Cre-expressing mouse strains behave identical to wild-type mice in regards to the parameters studied.

Response: *ShhCre*+/- Allele: Below is a list of references in which the Shhtm1(EGFP/cre)Cjt/J; from Harfe et al. was used in studies of urothelial development, regeneration or homeostasis. In addition, we have performed new experiments analyzing the urothelium in additional *ShhCre*+/- mice during development, homeostasis and after UTI (Figure 4).

The *ShhCre*+/- line has been used in a large number of studies including those listed below which specifically examine the bladder and urothelium. To our knowledge, no abnormalities have been identified in these heterozygotes. The line was generated by the Tabin lab. A quote from the original paper: "Evidence for an Expansion-Based Temporal *Shh* Gradient in Specifying Vertebrate Digit Identities. Brian D.Harfe, Paul J.Scherz, Sahar Nissim, Hua Tian, Andrew P.McMahon, Clifford J.Tabin. (2004). Cell 118, p517-528"

"In an effort to fate map cells that have expressed *Shh* in the mouse limb, we used gene targeting to insert a gene that encodes a *gfpcr* fusion protein into the *Shh* locus (see Experimental Procedures). The *gfpcr* cassette contained a nuclear localization signal and was inserted at the ATG of *Shh*. In addition, during the construction of the *Shhgfpcr* allele, the first 12 amino acids of *Shh* were removed to create a *Shh* null allele. ES cells in which the *gfpcr* cassette was correctly targeted were used to make mice, and these animals were then analyzed. **Mice heterozygous for the *Shhgfpcr* allele exhibited no noticeable phenotypes. *Shhgfpcr* heterozygous animals were viable and mated and produced offspring in expected Mendelian ratios.**"

Other studies using the *ShhCre*+/- line (Shhtm1(EGFP/cre)Cjt/J; from Harfe et al)

1. Kruppel-like factor 5 is required for formation and differentiation of the bladder urothelium. Sheila M.Bell, LiqianZhang, Angela Mendell, Yan Xu, Hans Michael Haitchi, James L. Lessard and Jeffrey A.Whitsett. (2011). Dev Biol. 358, p79-90.

Klf5^{flox/flox}Shh^{GfpCre-}, *Klf5^{flox/wt}Shh^{GfpCre-}*, and *Klf5^{flox/wt}Shh^{GfpCre+}* littermates were used as the controls. Urothelial differentiation in mutants and controls was examined, by immunostaining, RTPCR and microarray analysis. There were no noted differences between control groups.

2. Stage- and subunit-specific functions of polycomb repressive complex 2 in bladder urothelial formation and regeneration Chunming Guo, Zarine R. Balsara, Warren G. Hill, Xue Li. (2017). Development 144, p400-408.

Eed^{fl/+};ShhGC⁺ or *Ezh2^{fl/+};ShhGC⁺* mice were used as controls in this paper. The authors analyzed the developing and regenerating urothelium in *Eed* and *Eed/Ezh2* mutants. *In situ* hybridization revealed robust expression of *Shh*, *Ptched1* and *Ptched2* in controls. In a series of experiments, were no evident defects in urothelial development or regeneration in control mice.

3. In vivo replacement of damaged bladder urothelium by Wolffian duct epithelial cells. Diya B. Joseph, Anoop S. Chandrashekar, Lisa L. Abler, Li-Fang Chu, James A. Thomson, Cathy Mendelsohn, and Chad M. Vezina. (2018). PNAS 115, p8394-8399.

Shhcre^{+/+}; Dnmt1 flox^{+/+}; R26R^{+/+} embryos were used as controls in this study compared with age-matched mutant littermates (Shhcre^{+/+}; Dnmt1 flox^{/flox}; R26R^{+/+}=Dnmt1cKO). Analysis includes bladder urothelium and Wolffian duct, reveals apparently normal differentiation in controls.

4. Sonic hedgehog controls growth of external genitalia by regulating cell cycle kinetics. Ashley W. Seifert, Zhengui Zheng, Brandi K. Ormerod, and Martin J. Cohn. Nat Commun. (2010). 1, p1–9.

ShhCre^{+/-} mice used as controls. The urothelium is close to the Genital Tubercle and is prominent in several figures where differentiation and morphology appear normal.

5. Cell lineage analysis demonstrates an endodermal origin of the distal urethra and perineum. Ashley W. Seifert, Brian D. Harfe and Martin J. Cohn. (2008). Dev Biol. 318, p143-52.

The authors used ShhCre^{+/-} mice to investigate the origin of lower urinary tract epithelia, including urothelium. They generated a ShhGFPcre;R26R LacZ strain for lineage studies, which reveals regular morphology in 3D analysis, and regular differentiation of the urothelium and other lower urinary tract epithelia.

Studies using the Krt5CreERT2 and Up2CreERT2 lines (from the Chambon lab and the Cordon-Cardo lab, respectively) which were used as controls:

1. Mode of Surgical Injury Influences the Source of Urothelial Progenitors during Bladder Defect Repair. Schäfer FM, Algarrahi K, Savarino A, Yang X, Seager C, Franck D, Costa K, Liu S, Logvinenko T, Adam R, Mauney JR. (2017). Stem Cell Reports. 9, p2005-2017.

The authors used the Krt5CreERT2 and the Up2CreERT2 lines which are the same lines as used in our studies to examine lineage and urothelial regeneration using different types of injury models in combination with Gfp-reporter lines. Based on the extensive immunostaining analysis in this paper, there were no observed anomalies in urothelium in these reporter lines.

2. Bladder cancers arise from distinct urothelial sub-populations. Van Batavia J, Yamany T, Molotkov A, Dan H, Mansukhani M, Batourina E, Schneider K, Oyon D, Dunlop M, Wu XR, Cordon-Cardo C. and Mendelsohn, C. (2014). Nat Cell Biol. 16, p982-991.

Figure 1 shows Characterization comparing wild type urothelium with urothelium from Krt5CreERT2;mTmG and Up2CreERT2;mTmG lines (the mTmG is a Rosa26 reporter) that were used in lineage analysis studies of carcinogen induced bladder cancer. There were no differences observed between controls.

Other concerns:

1) The role of PPAR γ in bladder cancer is very controversial, with numerous studies both claiming an oncogenic or anti-cancerous role for PPAR γ . This is not evident from the way the manuscript is currently written. The short paragraphs in the introduction, results, and discussion that touch upon this subject are vague (using terms like “associated with” and “regulate”) and cite few recent publications. The authors should develop this further and more clearly (in a neutral manner), and include recent studies (examples PMIDs 30651555, 30912275).

Response: We have included the following paragraph in the introduction: “Mapping of the mutational landscape of muscle-invasive bladder cancers (MIBC) together with unsupervised clustering analysis of the whole genome expression data revealed that MIBC can be sub-categorized into luminal and basal subtypes. These subtypes are histologically distinct and display discrete sets of mutations and gene expression signatures¹⁴⁻¹⁹. These analyses reveal alterations in *PPARG* expression and signaling, suggesting that *PPARG*-dependent transcriptional regulation may be important in the etiology of urothelial carcinoma. Supporting this, *PPARG* copy number expansion and increased expression of *FABP4*, a direct *PPARG* transcriptional target, were observed in luminal-tumors²⁰⁻²². Activating mutations in *PPARG* and *RXRA*, a *PPARG* binding partner, were also observed in luminal MIBC^{23,24}. In addition, upregulation of these gene sets important for lipid metabolism and adipogenesis in patients that harbor *PPARG* gain-of function mutations suggest that *PPARG* may be an important regulator of lipid metabolism in the luminal subtype of MIBC.

The exact contribution of *PPARG* to the etiology of the basal subtype of urothelial carcinoma is less clear. *PPARG* expression is low in basal subtype tumors compared to healthy urothelium, and *PPARG* is down-regulated in Claudin-low tumors, which have basal-like features. Interestingly, genes encoding cytokines and chemokines are up-regulated in Claudin-low basal-like tumors, which may reflect unregulated NF- κ B signaling due to low levels of *PPARG*²⁵. Expression of *PPARG* and its binding partner *RXRA*, are reduced in the SCC-like (SCCL) subtype of MIBC, which shares many features with the basal subtype, including gene expression signatures and common mutations. Transcriptional analysis of those basal or basal-like subtype of tumors revealed a large cluster of genes important for lipid metabolism that were down-regulated compared to the luminal subtype of tumors. *In silico* Chip-Seq analysis revealed that many of these down-regulated genes contained *PPARG* binding sites in their regulatory regions, suggesting that these are *PPARG*-transcriptional targets²⁶.”

We have included the following paragraph in the discussion:

“Positive and negative *Pparg* signaling can have profound effects on bladder cancer cells⁵⁶ and on immune functions in MIBC⁵⁷. *Pparg* expression is down-regulated in basal/squamous subtypes of MIBC compared to the healthy urothelium, and up-regulated in the luminal subtype of MIBC. We observed a number of abnormalities in *Pparg* mutants that are similar to those observed in bladder cancers with low *Pparg* expression, in particular the basal/squamous subtype of MIBC and in Claudin-low (basal-like) subtype tumors. Similarities include up-regulated expression of markers of basal differentiation (*Krt14*, *Krt6*, *Krt5*) and up-regulation of NF- κ B and expression of inflammatory molecules^{25,58}. We also observed alterations in the urothelium that are associated with tumor formation, including up-regulation of *Snail1*, *Slug* and *Vimentin*, and a compromised basement membrane (Fig. 5). Despite these similarities, we did not observe invasion or signs of tumor formation in *ShhCre;Pparg^{fl/fl}* mutants, suggesting that *Pparg* mutations are unlikely to be primary drivers of tumor formation. We believe future studies should focus on whether *Pparg* mutations promote tumorigenesis in cooperation with other mutations, or whether *Pparg* acts down-stream regulating cellular differentiation or immune functions after tumor initiation.”

2) In the results (page 10), it is stated that “*Upk3*, which labels I-cells and S-cells, and *Krt20* which labels mature S-cells were both down-regulated (Fig. 3a-i)”. However, Fig. 3i shows an increase in S cells. Please explain.

Response: The problem here is that we have included numbers of immature S-cells in the analysis of S-cell numbers, which is not informative, since the main point is that luminal cells are present in mutants but have down-regulated S-cell markers. We have eliminated the top panel graph in Fig. 3i, which we agree, is confusing, along with the associated text, which has been changed to: “This analysis also revealed that *Upk3a*, which is highly enriched in S-cells, and *Krt20*, which labels mature S-cells, were both down-regulated (Fig. 3a-h; Supplementary Fig. 3a-f), suggesting that *Pparg*-dependent transcription is important for proper maintenance of S-cells. Unexpectedly, we observed *Ki67* expression in 20% of S-cells and 10% of I-cells in *Upk2CreERT2;Pparg^{fl/fl}* mutants during homeostasis (Fig. 3c,g,i).”

3) *PPAR*gamma can regulate NFkappaB through different mechanisms, including by sequestering the P65 subunit, but also by regulating I-kappa-B-alpha and *IKK*-alpha/beta levels. It is not clear what is stained as NFkappaB in Fig. 5 and Supplemental Fig. 4 (no description in Methods section). The whole part on NFkappaB should be developed in more detail, including mechanistic details.

Response: We have included the following paragraph in the discussion section:

Pparg is known to be an important regulator of inflammatory response, in part by regulating transcriptional activity of NF- κ B, which among other things, controls the innate immune response to UPEC infection. *p65/Rela*, one of five NF- κ B family members, is transiently up-regulated in the wild type urothelium in response to UPEC infection, but persists in *ShhCre;Pparg^{fl/fl}* mutants for months accompanied by edema and leukocyte infiltration (Fig 5), suggesting that mutants fail to resolve the NF- κ B response. NF- κ B is a transcription factor that regulates expression of immune genes, and also plays an important role in epithelial barriers such as the skin, gut and esophagus, controlling recognition and response to invading pathogens⁴⁷⁻⁴⁹. The NF- κ B family has 5 members, including *p65/Rela*, which contains a transactivation domain that can bind to DNA and positively regulate transcription. NF- κ B signaling is mediated by heterodimers in the urothelium (most likely including *p65/Rela*) that are sequestered in the cytoplasm in an inactive state in resting cells, where they are bound to the inhibitor *Ikb*. Stimulus induces degradation of *Ikb* and release of NF- κ B-heterodimers, which move to the nucleus and bind to response elements in target genes, activating their transcription⁵⁰⁻⁵². NF- κ B signaling, as evidenced by *p65* expression, is activated rapidly in the wild type urothelium in response to UPEC infection. Recent studies suggest that its initial activation may be triggered by binding of the bacterial adhesin, *Fimh*, to *Upk1a*, which is

expressed on the surface of S-cells. This interaction triggers TLR4 mediated pattern recognition⁵³. *Pparg* is known to be an important regulator of NF- κ B-transcriptional activity, acting as a transrepressor. *Pparg* can bind directly to the NF- κ B protein, preventing its interaction with promoter regions of target genes. SUMOylation of the *Pparg* ligand binding domain enables *Pparg* to bind to regulatory sequences of NF- κ B target genes, preventing disassociation of co-repressors and resulting in repression of NF- κ B-dependent transcription^{54,55}. While the direct mechanism by which *Pparg* controls NF- κ B in the urothelium is unclear, our studies provide strong evidence that *Pparg* regulates *p65/Rela* expression and is required in urothelial cells to suppress the innate immune response induced by UPEC infection.

4) Supplement Fig. 3 is never mentioned in the text.

Response: Please see response to point 2, including a reference to Supplementary Figure 3 which has been added to the text.

The manuscript has been written in a very sloppy manner and includes numerous errors of which a non-exhaustive list follows below. Especially the Figure legends contained many errors and the Methods section is far from complete. Even though these are minor concerns, the authors could have made a bigger effort to submit a version that doesn't come across as an early draft.

1) Introduction (page 5): “*Pparg* expression is down-regulated in the basal subtype of BC, which has squamous features”. It is not clear what BC refers to; bladder cancer or basal cells.

Response: Paragraph has been replaced (see above, point 1)

2) Results (page 7): “*Pparg* expression is undetectable in the Basal cell compartment until E18 (Fig. 1g, green arrow).” Fig. 1g shows situation at E16, not E18.

Response: This is a typo; basal cells are detectable at E16. Sentence now reads: “*Pparg* expression is undetectable in the Basal cell compartment until **E16** (Fig. 1g, green arrow)”

3)Page 9: “However, the mitochondrial membrane is impermeable to fatty acids and a specialized carnitine carrier system consisting of *Cpt1*, *Slc25a20* and *Cpt2* controls fatty acid transport (Spinelli and Haigis, 2018). We observed down-regulation of all 3 genes in *ShhCre;Ppargfl/fl* mutants (Fig. 2b,e). In addition, 15 genes that encode members of complex 1 NADH ubiquinone oxidoreductase were down-regulated, including *Nd4*, *Nd5* and *Nd6* that are transcribed from mitochondrial DNA (Fig. 2b; *Cpt2* immunostaining is shown in Fig. 2e)”

It would make more sense to write:

“However, the mitochondrial membrane is impermeable to fatty acids and a specialized carnitine carrier system consisting of *Cpt1*, *Slc25a20* and *Cpt2* controls fatty acid transport (Spinelli and Haigis, 2018). We observed down-regulation of all 3 genes in *ShhCre;Ppargfl/fl* mutants (Fig. 2b, *Cpt2* immunostaining is shown in Fig. 2e). In addition, 15 genes that encode members of complex 1 NADH ubiquinone oxidoreductase were down-regulated, including *Nd4*, *Nd5* and *Nd6* that are transcribed from mitochondrial DNA (Fig. 2b)”

Response: The original paragraph has been replaced with the suggested paragraph.

We have corrected the following errors in the text:

4) Page 9: “Fig. S2 b,c” Be consistent and write Supplementary Fig. 2 b,c

5) Page 10: “cell population in *ShhCre;Ppargfl/fl* mutants (Fig. 1h,i,k,o; Supplementary Fig. 1k-n).” Should be “Fig. 1h,i,k,o”.

6) Page 12: “an possibility” should be “a possibility”.

7) Page 12: “may normally act in basal cells suppress squamous” should be “basal cells to suppress”.

8) Page 13: All referrals to Supplementary Fig. 5 in first paragraph should actually refer to Supplementary Fig. 4.

9) Page 13: “phenotype that continuing after infection was cleared (Fig. 5a-i)” should be “phenotype that continued after infection was cleared (Fig. 5f-h,i)”.

10) The Methods section does not describe the mTmG mice used for Supplementary Fig. 5. Furthermore, the section on immunostaining only lists a fraction of the primary antibodies used for the stainings shown. Missing are PPARgamma, FABP4, Krt14, CD19, CD3, F4/80, CD45, NFkappaB, SOD1/2, Cpt2, Cox1, etc....

Response: Source of mTmG line and primers used for genotyping have been added to the methods section.

11) Figure 2 legend: “(f-h) Immunofluorescence staining of superficial cell markers (f) Cldn8, (g) Sprr1a and (h) Uchl1” should be “(i-k) Immunofluorescence staining of superficial cell markers (i) Cldn8, (j) Sprr1a and (k) Uchl1”

Response: Fixed.

12) Supplement Fig. 2 legend: arrows indicating vesicles are not mentioned.

Response: The red arrow in (c) points to the abnormal tight junction in a mutant S-cell. Green arrowheads in (b) point to fusiform vesicles in controls. Red arrowheads point to abnormal vesicles in the mutant (c); added to text of Figure legend.

13) Figure 3 legend: “Co-staining of (a, e) Krt20 and P63, (b, f) Fabp4 and Upk3a, (c, g) Ki67 and Upk3a, (d, h)” should be “Co-staining of (a, e) Krt20 and PPARgamma, (b, f) Fabp4, P63, and Upk3a, (c, g) Ki67, P63, and Upk3a, (d, h)”.

Response: Fixed

14) Figure 3 legend: “numbered from the smallest inclusion (1) to the largest inclusion (4).” No such numbering is visible in the figure.

Response: “numbered from the smallest inclusion (1) to the largest inclusion (4).” removed

15) Figure 4 legend: “(a-d) Transmission electron microscopy image of the urothelium from adult *Pparg*^{fl/fl} controls or *Krt5CreER*;*Ppar* fl/fl mutants.” Should be “*ShhCreER*;*Ppar* fl/fl”

Response: Transmission electron microscopy image of the urothelium from adult *Pparg*^{fl/fl} controls or *ShhCre*;*Ppar*^{fl/fl} mutants.

16) Figure 5 legend: Panel i is not mentioned/described.
All those should be easy to edit.

Response: Added to figure legend: “(i) Schematic representation of the urothelium in controls and in *ShhCre*;*Pparg*^{fl/fl} mice 72h after UTI”.

Reviewers' Comments:

Reviewer #1:

Remarks to the Author:

No additional comments, data, or edits requested.

Reviewer #2:

Remarks to the Author:

The authors have addressed all my concerns.

Reviewer #3:

Remarks to the Author:

In this revised manuscript the authors have dealt with most concerns in a satisfactory manner. However some minor concerns are still remaining:

1) Introduction, second paragraph: "These binucleated I-cells than double their and differentiate into S-cells." What does "their" refer to?

2) Introduction, last paragraph: "...and prevents basal cells from undergoing squamous differentiation and is." End of sentence is missing.

3) Fig. 1a is never mentioned in the results section. Neither is Fig. 5i.

4) Results, last sentence first paragraph: "...failed to self-renew in or regressed". Not sure what "in" does in this sentence.

5) Results, page 7, last paragraph: "including the complement cascade". This data is not part of the manuscript. Similarly, legend of Fig. 2a refers to heatmaps of genes related to complement cascade and innate immunity. Both are not part of this figure.

6) Results, page 9: "Slug a marker expressed cells undergoing EMT". Should say: "Slug, a marker expressed in cells undergoing EMT". Furthermore, EMT should be defined.

7) Results, page 10, first paragraph: "...by 72 hous" should be "by 72 hours"

8) The cell tracing/reporter experiment using mTmG mice, shown in Supplementary Fig. 5c', is still not adequately described in the manuscript. The authors now only mention the mouse line in the methods but don't describe the actual experiment in the Methods, Results, and Figure Legend. They should explain that the ShhCre mice were crossed with the mTmG mice. As a result, cells that have Cre activity are green (GFP) and cells that don't are red (Tomato). The legend for Supplementary Fig. 5c' wrongly states that cells were harvested from ShhCre; Ppargfl/fl mutants. The flow cytometry experiment shown in Supplementary Fig. 5c' is not described in the Methods section.

9) Discussion, first sentence: "In this study, show that.." should be "In this study, we show that...".

10) Discussion, second paragraph: "There are 4 Ppar family members..." There are only 3 members; Pparb/d is one gene for which the official nomenclature is Ppard.

11) Figure Legends, Supplement Fig. 1: Refers to panel o that should show data on measurement

of membrane integrity. These measurements are also mentioned in the response to major concern 3 of Reviewer 2. However, this panel is lacking.

12) Legend Fig. 3, panel i: "Quantification of the proportion of different cell types" should be "Quantification of the proportion of Ki67+ cells of different cell types".

13) Legend Fig. 5, panels a' and b': should specify that its 4 wks after UTI. The panel b' of the figure should specify the mouse strains above the heat maps.

14) Legend Supplementary Fig. 5: There are two panels g. Furthermore, it should be mentioned at what time the control mice shown in panel g and k were sacrificed; at 2 wks, 5 months, or 1 year.

Lastly, I would like to mention that most, if not all, of these concerns are due to lack of adequate proofreading by the authors and could have easily been avoided. In the rebuttal there is further evidence of this lack of rigor in the form of a sentence at the end: "All those should be easy to edit". The latter most likely being a comment from one of the authors that was left.

Reviewer #3 (Remarks to the Author):

In this revised manuscript the authors have dealt with most concerns in a satisfactory manner. However some minor concerns are still remaining:

1) Introduction, second paragraph: "These binucleated I-cells than double their and differentiate into S-cells." What does "their" refer to?

Response: Sentence now reads: "These binucleated I-cells undergo a second round of endoreplication differentiating into S-cells with $4n/4n$ ploidy⁴"

2) Introduction, last paragraph: "...and prevents basal cells from undergoing squamous differentiation and is." End of sentence is missing.

Response: Sentence now reads: "Pparg plays an independent role in basal cells, preventing squamous-like differentiation."

3) Fig. 1a is never mentioned in the results section. Neither is Fig. 5i.

Response: 1a is now included in the text describing Fig. 1.

4) Results, last sentence first paragraph: "...failed to self-renew in or regressed". Not sure what "in" does in this sentence.

Response: corrected

5) Results, page 7, last paragraph: "including the complement cascade". This data is not part of the manuscript. Similarly, legend of Fig. 2a refers to heatmaps of genes related to complement cascade and innate immunity. Both are not part of this figure.

Response: These are part of Figure 5 (ORA) and the text has been changed to reflect this.

6) Results, page 9: "Slug a marker expressed cells undergoing EMT". Should say: "Slug, a marker expressed in cells undergoing EMT". Furthermore, EMT should be defined.

Done

7) Results, page 10, first paragraph: "...by 72 hous" should be "by 72 hours"

Done

8) The cell tracing/reporter experiment using mTmG mice, shown in Supplementary Fig. 5c', is still not adequately described in the manuscript. The authors now only mention the mouse line in the methods but don't describe the actual experiment in the Methods, Results, and Figure Legend. They should explain that the ShhCre mice were crossed with the mTmG mice. As a result, cells that have Cre activity are green (GFP) and cells that don't are red (Tomato).

Response: Included in legend to Supplementary Figure 7: "Pparg is deleted in urothelial cells of ShhCre;Ppargfl/fl mutants not in immune cells. (a) To confirm that the ShhCre driver is not active in immune cells, ShhCre;Ppargfl/fl mice were crossed with mTmGfl/fl (Gt(ROSA)26Sortm4(ACTB-tdTomato,-EGFP)Luo/J) mice (hereafter referred to as mTmG mice) to generate ShhCre;Ppargfl/fl;mTmG mice. In this line, cells undergoing Cre-dependent recombination will express Gfp, and cells that don't undergo recombination will express mTomato. Recombination in the urothelium and in immune cells was assessed by FACS analysis of urothelial cells and immune cells harvested from ShhCre;Ppargfl/fl;mTmG mutants 24hrs post infection."

The legend for Supplementary Fig. 5c' wrongly states that cells were harvested from ShhCre;Ppargfl/fl mutants.

Done

The flow cytometry experiment shown in Supplementary Fig. 5c' is not described in the Methods section.

Response: It is now included in Methods Section: " Analysis of Cre-dependent recombination in leukocytes"

9) Discussion, first sentence: "In this study, show that.." should be "In this study, we show that..".

Response-Changed to: "In this study, we show that *Pparg* plays an essential role as a regulator of urothelial development in vivo, controlling differentiation and/or survival of basal cells, I-cells and S-cells.

10) Discussion, second paragraph: "There are 4 Ppar family members..." There are only 3 members; Pparb/d is one gene for which the official nomenclature is Ppard.

Response: changed to 3

11) Figure Legends, Supplement Fig. 1: Refers to panel o that should show data on measurement of membrane integrity. These measurements are also mentioned in the response to major concern 3 of Reviewer 2. However, this panel is lacking.

Response to 11)

from the Rebuttal: Reviewer 2: *"While it is almost certainly safe to conclude (based on the findings) that Pparg regulates mitochondrial function in a cell autonomous manner, I am not sure that the same can be said about the cell autonomous manner in which pparg regulates cell differentiation or survival. This is true because it is possible that Pparg ablation influences barrier integrity, resulting in urine leaking through the urothelium, and potentially adversely affecting cell differentiation or survival. Therefore, this conclusion should be scaled back a bit.*

from the rebuttal: Response to reviewer 2: *"We have not observed differences in permeability between mutants and controls using methylene blue staining, but we have not performed more exhaustive assays of permeability; hence we will tone down the text as follows: The title of the section "Pparg regulates differentiation, mitochondrial functions and survival of S-cells in a cell autonomous manner" is now: "Pparg regulates mitochondrial functions in S-cells."*

The figure in question was Supplementary Figure 1o. It was included as a response to a major concern of Reviewer 2. But we were not asked to perform a direct assessment of membrane permeability, we were asked to tone down the message regarding cell autonomous role of Pparg, which we did. We did not intend to include the figure in the manuscript, this was a mistake, it was meant to be included only in the rebuttal letter, hence it has been removed. We do not think that this data provides strong enough evidence on its own to rule out membrane permeability.

12) Legend Fig. 3, panel i: "Quantification of the proportion of different cell types" should be "Quantification of the proportion of Ki67+ cells of different cell types".

Done

13) Legend Fig. 5, panels a' and b': should specify that its 4 wks after UTI. The panel b' of the figure should specify the mouse strains above the heat maps.

Done

14) Legend Supplementary Fig. 5: There are two panels g. Furthermore, it should be mentioned at what time the control mice shown in panel g and k were sacrificed; at 2 wks, 5 months, or 1 year.

Done